# Grasping the Arrow of Time from the Singularity: Decoding Micromotion in Low-dimensional Latent Spaces from StyleGAN

## Abstract

The disentanglement of StyleGAN latent space has paved the way for realistic and controllable image editing, but does StyleGAN know anything about temporal motion, as it was only trained on static images? To study the motion features in the latent space of StyleGAN, in this paper, we hypothesize and demonstrate that a series of meaningful, natural, and versatile small, local movements (referred to as "micromotion", such as expression, head movement, and aging effect) can be represented in low-rank spaces extracted from the latent space of a conventionally pre-trained StyleGAN-v2 model for face generation, with the guidance of proper "anchors" in the form of either short text or video clips. Starting from one target face image, with the editing direction decoded from the low-rank space, its micromotion features can be represented as simple as an affine transformation over its latent feature. Perhaps more surprisingly, such micromotion subspace, even learned from just single target face, can be painlessly transferred to other unseen face images, even those from vastly different domains (such as oil painting, cartoon, and sculpture faces). It demonstrates that the local feature geometry corresponding to one type of micromotion is aligned across different face subjects, and hence that StyleGAN-v2 is indeed "secretly" aware of the subject-disentangled feature variations caused by that micromotion. We present various successful examples of applying our low-dimensional micromotion subspace technique to directly and effortlessly manipulate faces. Compared with previous methods, our framework shows high robustness, low computational overhead, and impressive domain transferability. Our code will be released upon acceptance.

## 1 Introduction

In recent years, the StyleGAN and its variants (Karras et al., 2018; 2021; 2019; 2020; Sauer et al., 2022) have achieved state-of-the-art performance in controllable image synthesis. These high qualities and fine-grained controls of the synthesized images are largely associated with the expressive latent space of StyleGAN. Previous research has revealed that the learned latent space of StyleGAN can be smooth and interpretable (Abdal et al., 2019; 2020; Wu et al., 2021; Zhu et al., 2020). Furthermore, previous studies (Karras et al., 2019; 2020) have shown that by feature manipulations and interpolations in the latent space, the style-based GANs can generate a variety of intriguing images with desired changes. These findings have led to many downstream applications such as face manipulation (Wei et al., 2021; Alaluf et al., 2021a), style transfer (Abdal et al., 2019; Kwon & Ye, 2021), general image editing (Gu et al., 2020; Park et al., 2020; Suzuki et al., 2018), and even video generation (Chu et al., 2020; Fox et al., 2021; Skorokhodov et al., 2021; Zhang & Pollett, 2021).

Given this phenomenal result, many try to further understand the potential in the latent space of Style-GAN. Particularly, rather than per-image editing methods, people wonder whether it is possible to directly locate a series of latent codes that correspond to sample-agnostic semantically meaningful attributes (*e.g.* smiling, aging on human faces). These attempts to disentangle meaningful editing directions can be roughly categorized into supervised methods and unsupervised methods. The supervised approaches (Shen et al.,

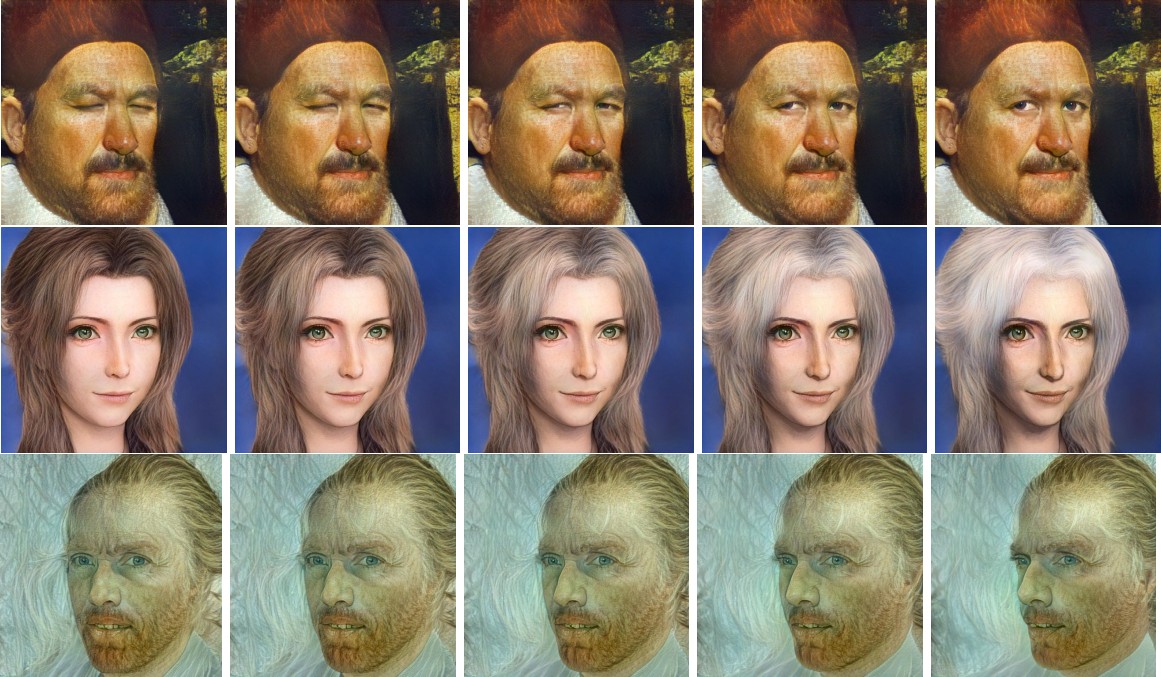

Figure 1: **Representative examples created by the proposed method.** The original images are edited using a simple linear scaling with the discovered universal editing directions on various transformations. These three rows correspond to *eye-opening, aging, and head rotation.*

2020b; Yang et al., 2021; Goetschalckx et al., 2019) typically sample a series of latent codes, labeling the latent codes with pretrained attributes predictors, and finally learning classifiers for each desired attribute in the latent space. On the other hand, the unsupervised approach (Shen & Zhou, 2021; Härkönen et al., 2020) explores the principal components of the sampled latent codes and observes if these codes correspond to semantically meaningful editing directions. However, as will be shown later, the editing directions found by these methods are shown to be still entangled with other attributes. When applying these discovered editing directions, the result images suffer from undesired changes in identity and other attributes. This leads to the following question: whether this sub-optimal entanglement is due to the intrinsic limits of the entangled latent space, or it is because previous methods do not fully reveal the potentials of the StyleGAN?

To answer the question, in this paper, we propose in-depth investigations on the StyleGAN-v2's latent space trained on face generations. In particular, we hypothesize that from the StyleGAN's high dimensional latent space, a low-rank feature space can be extracted where universal editing directions can be reconstructed for various facial style transformations including changes in expressions/emotions, head movements, and aging effects, which we refer to as a series of **micromotions**. Thanks to the highly disentangled essence of the decoded editing directions, for any given input, linear scaling along the same found direction will make the image change its style smoothly. Furthermore, to find such a directional vector we leverage the guidance of proper "anchors" in the form of either short texts or a reference video clip and show the directional vector can be efficiently found via simple subtractions using a robustly learned linear subspace projection. Surprisingly, such latent subspace can be extracted using only a single query image, and then the resulting editing direction can be used for any unseen face image, even for those from vastly different domains including oil painting, cartoon, sculpture, *etc.* Figure 1 shows the generated images for multiple style transformations and face types. The contributions of our paper are three-fold:

- Leveraging the low-dimensional feature space hypothesis, we demonstrate the properties of StyleGAN's latent space from a global and universal viewpoint, using "micromotions" as the subject.

- We demonstrate that by using text/video-based anchors, low-dimensional micromotion subspace along with universal and highly disentangled editing directions can be consistently discovered using the same robust subspace projection technique for a large range of micromotion-style facial transformations.

- We show the editing direction can be found using a single query face input and then directly applied to other faces, even from vastly different domains (*e.g.*, oil painting, cartoon, and sculpture faces), in an easily controllable way as simple as linear scaling along the discovered subspace.

## 2 Related Works

### 2.1 StyleGAN: Models and Characteristics

The StyleGAN (Karras et al., 2021; 2019; 2020) is a style-based generator architecture targeting image synthesis tasks. Leveraging a mapping network and affine transformation to render abstract style information, StyleGAN is able to control the image synthesis in a scale-specific fashion. Particularly, by augmenting the learned feature space and hierarchically feeding latent codes at each layer of the generator architecture, the StyleGAN has demonstrated surprising image synthesis performance with controls from coarse properties to fine-grained characteristics (Karras et al., 2019). Also, when trained on a high-resolution facial dataset (*e.g.,* FFHQ (Karras et al., 2019)), the StyleGAN is able to generate high-quality human faces with good fidelity.

### 2.2 StyleGAN-based Editing

Leveraging the expressive latent space by StyleGAN, recent studies consider interpolating and mixing the latent style codes to achieve specific attribute editing without impairing other attributes (e.g. person identity). (Hou et al., 2022; Shen et al., 2020a; Tewari et al., 2020a;b; Wu et al., 2021) focus on searching latent space to find latent codes corresponding to global meaningful manipulations, while (Chong et al., 2021) utilizes semantic segmentation maps to locate and mix certain positions of style codes to achieve editing goals.

To achieve zero-shot and open-vocabulary editing, recent works set their sights on using pretrained multi-modality models as guidance. With the aligned image-text representation learned by CLIP, a few works (Wei et al., 2021; Patashnik et al., 2021) use text to extract the latent edit directions with textual defined semantic meanings for separate input images. These works focus on extracting latent directions using contrastive CLIP loss to conduct image manipulation tasks such as face editing (Patashnik et al., 2021; Wei et al., 2021), cars editing (Abdal et al., 2021a). Besides, a few recent works manipulate the images with visual guidance (Lewis et al., 2021; Kim et al., 2021). In these works, image editing is done by inverting the referential images into corresponding latent codes, and interpolating the latent codes to generate mixed-style images. However, these works focus on per-example image editing. In other words, for each individual image input, they have to compute corresponding manipulations in the latent space separately. With the help of disentangled latent space, it is interesting to ask whether we can decode universal latent manipulations and conduct sample-agnostic feature transformations.

### 2.3 Feature Disentanglement in Latent Space of StyleGAN

Feature disentanglement in StyleGAN latent space refers to decomposing latent vector components corresponding to interpretable attributes. Previous studies on StyleGAN latent space disentanglement can be roughly categorized into supervised and unsupervised methods. In supervised methods (Shen et al., 2020b; Yang et al., 2021; Goetschalckx et al., 2019), they typically leverage auxiliary classifiers or assessors to find the editing directions. To be more specific, Shen et al. (2020b) first sample a series of latent codes from the latent space and render corresponding images. Then, they train an SVM to learn the mapping between sampled latent codes and corresponding attributes, where the labeled attributes are supervised by the auxiliary classifiers. Finally, the normal direction of the hyperplane is the found editing direction. Goetschalckx et al. (2019) directly optimize the editing direction based on an auxiliary classifier. On the other hand, the unsupervised methods typically explore the principal components of sampled latent codes, while they manually check if these components correspond to semantically meaningful attributes. However, as will be shown later, the editing directions found by these methods are shown to be still entangled with other attributes. In this work, leveraging a stronger low-rank latent space hypothesis, we find highly-disentangled latent codes and show that sample-agnostic editing directions can be consistently found in StyleGAN's latent space.

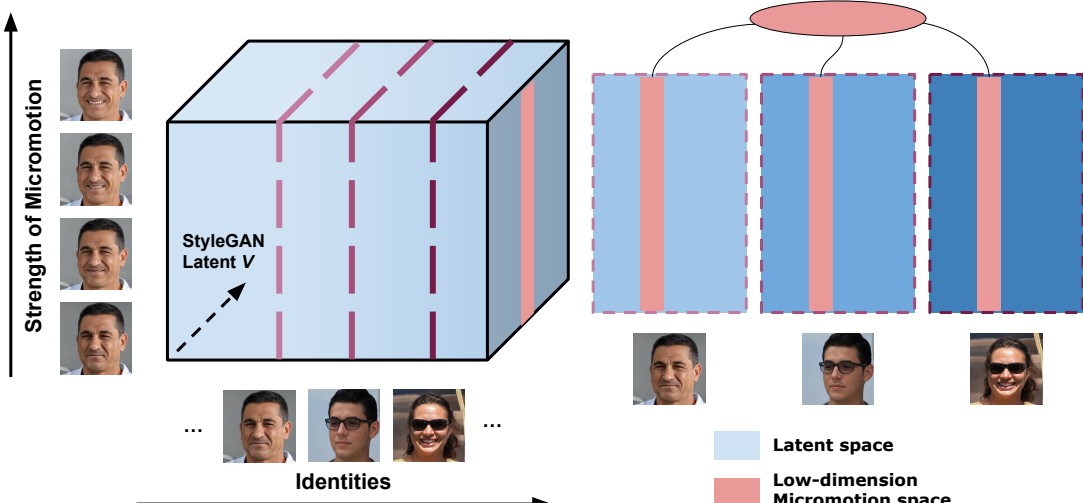

Figure 2: **A tensor illustration of our hypothesis.** In the StyleGAN latent space, we hypothesize the same type of micromotion, *at different quantitative levels but for the same identity*, can be approximated by a low-rank subspace. We further hypothesize that subspaces *for the same type of micromotion found at different identities* are extremely similar to each other, and can hence be transferred across identities.

## 3 Method

In this section, we first present the problem of decoding micromotion in a pre-trained StyleGAN latent space, and we define the notations involved in this paper. We then articulate the low-rank micromotion subspace hypothesis in Sec. 3.2, proposing that the locally low-dimensional geometry corresponding to one type of micromotion is consistently aligned across different face subjects, which serves as the key to decode universal micromotion from even a single identity. Finally, based on the hypothesis, we demonstrate a simple workflow to decode micromotions and seamlessly apply them to various in-domain and out-domain identities, incurring clear desired facial micromotions.

### 3.1 Problem Setting

Micromotions are reflected as smooth transitions in continuous video frames. In a general facial-style micromotion synthesis problem, given an arbitrary input image $I_0$ and a desired micromotion (*e.g.* smile), the goal is to design an identity-agnostic workflow to synthesize temporal frames $\{I_1, I_2, \ldots, I_t\}$, which constitute a consecutive video with the desired micromotion.

Synthesizing images with StyleGAN requires finding proper latent codes in its feature space. We use $G$ and $E$ to denote the pre-trained StyleGAN synthesis network and StyleGAN encoder respectively. Given a latent code $\mathbf{V} \in \mathcal{W}^+$, the pre-trained generator $G$ maps it to the image space by $I = G(\mathbf{V})$. Inversely, the encoder maps the image $I$ back to the latent space $\mathcal{W}^+$, or $\hat{\mathbf{V}} = E(I)$. Leveraging the StyleGAN latent space, finding consecutive video frames turns out to be a task of finding a series of latent codes $\{\mathbf{V}_1, \mathbf{V}_2, \ldots, \mathbf{V}_t\}$ corresponding to the micromotion.

### 3.2 Key Hypothesis: The Low-rank Micromotion Subspace

To generate semantically meaningful and correct micromotions using StyleGAN, the key objective is to find proper latent code series in its feature space. We hypothesize that those latent codes can be decoded by a low-rank micromotion subspace. Specifically, we articulate the key hypothesis in this work, stated as: *The versatile facial style micromotions can be represented as low-rank subspaces within the StyleGAN latent space, and such subspaces are subject-agnostic.*

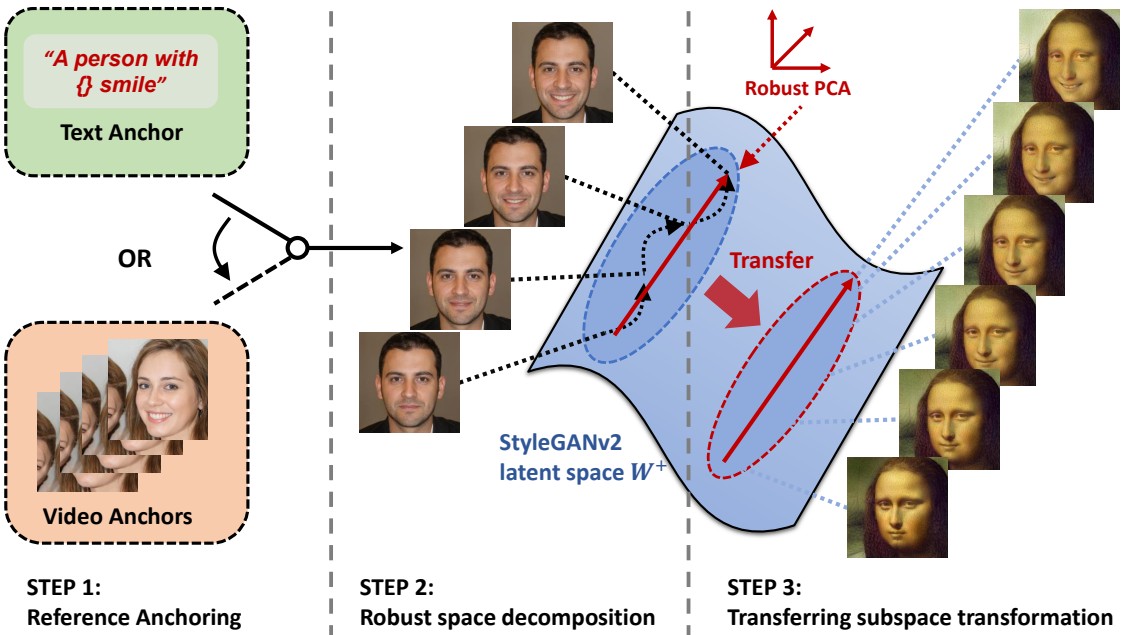

Figure 3: **Our workflow illustration.** In our workflow, we first extract a low-dimensional micromotion subspace from one identity, and then transfer it to a novel identity "Mona-lisa".

To give a concrete illustration of the hypothesis, we plot a tensor-view illustration of a micromotion subspace, smile, in Figure 2. The horizontal axis encodes the different face identities, and each perpendicular slice of the vertical plane represents all variations embedded in the StyleGAN latent space for a specific identity. We use the vertical axis to indicate the quantitative strength for a micromotion (*e.g.*, smile from mild to extreme). Given a sampled set of images in which a subject face changes from the beginning (e.g., neutral) to the terminal state of a micromotion, each image can be synthesized using a latent code $\mathbf{V}$. Aligning these latent codes for one single subject formulates a *micromotion matrix* with dimension $V \times M$, where $V$ is the dimension of the latent codes and $M$ is the total number of images. Eventually, different subjects could all formulate their micromotion matrices in the same way, yielding a *micromotion tensor*, with dimension $P \times V \times M$ assuming a total of $P$ identities. Our hypothesis is then stated in two parts:

- Each subject's micromotion matrix can be approximated by a "micromotion subspace" and it is inherently low-rank. The micromotion "strengths" can be reduced to linearly scaling along the subspace.

- The micromotion subspaces found at different subjects are substantially similar and even mutually transferable. In other words, different subjects (approximately) share the common micromotion subspace. That implies the existence of universal edit direction for specific micromotions regardless of identities.

If the hypothesis can be proven true, it would be immediately appealing for sample-agnostic image manipulations. First, micromotion can be represented in low-dimensional disentangled spaces, and the dynamic edit direction can be reconstructed once the space is anchored. Second, when the low-dimensional space is found, it can immediately be applied to multiple other identities with extremely low overhead, and is highly controllable through interpolation and extrapolation by as simple as linear scaling.

## 3.3 Our Workflow

With this hypothesis, we design a workflow to extract the edit direction from decomposed low-dimensional micromotion subspace, illustrated in Figure 3. Our complete workflow can be distilled down to three simple steps: (a) collecting anchor latent codes from a single identity; (b) enforcing robustness linear decomposition to obtain a noise-free low-dimensional space; (c) applying the extracted edit direction from low-dimensional space to arbitrary input identities.

**Step 1: Reference Anchoring.** To find the edit direction of a specific micromotion, we first acquire a set of latent codes corresponding to the desired action performed by the same person. Serving as anchors, these latent codes help to disentangle desired micromotions in later steps. Here, we consider two approaches, text-anchored and video-anchored methods, respectively.

Text-anchored Reference Generation: The StyleCLIP (Patashnik et al., 2021) has shown that expressive phrases can successfully manipulate attributes in synthesized images. We leverage the its latent optimization pipeline to generate the anchoring latent codes for desired micromotions. The main-idea to optimize these latent codes is to minimize the contrastive loss between the designed input texts and the images rendered by the codes with a few regularizations. Here, one major question is how to design the most appropriate text template to guide the optimization. To generate images with only variance in degrees of micromotions, a natural method is to specify the degrees in the text, where we concatenate a series of adjectives or percentages with the micromotion description text to indicate the various strength and the stage of the current micromotion. For example, for the micromotion "eyes closed", we use both percentages and adjectives to modify the micromotion by specifying "eyes *greatly/slightly* closed" and "eyes *10%/30%* closed". Here, we emphasize that this is just one of the possible text prompts design options. We compare various choices of text prompts, and the experiments of the text prompt choices will be covered in the ablation study.

Video-anchored Reference Generation: StyleCLIP relies on text guidance to optimize the latent codes, while for abstract and complicated motions, such as a gradual head movement with various head postures, the text might not be able to express the micromotion concisely. To overcome this issue, we leverage a reference video demonstration to anchor the micromotion subspace instead. In the reference video-based anchoring methods, we use frames of reference videos to decode the desired micromotions. Specifically, given a reference video that consists of continuous frames, we invert these frames with a pre-trained StyleGAN encoder to obtain the reference latent codes. We emphasize that different from the per-frame editing method, the goal of using reference video frames is to anchor the low-dimensional micromotion subspace. Thus, we use much fewer frames than per-frame editing methods, and no further video frames are used once we extract the space.

After applying either anchoring method, we obtain a set of $t_n$ referential latent codes denoted as $\{\mathbf{V}_{t1}, \mathbf{V}_{t2}, \ldots, \mathbf{V}_{tn}\}$. We will use these codes to obtain a low-rank micromotion space in later steps.

**Step 2: Robust space decomposition.** Due to the randomness of the optimization and the complexity of image contents (e.g., background distractors), the latent codes from the previous step could be viewed as "noisy samples" from the underlying low-dimensional space. Therefore, based on our low-rank hypothesis, we leverage further decomposition methods to robustify the latent codes and their shared micromotion subspace.

The first simple decomposition method we adopt is the principal component analysis (PCA), where each anchoring latent code serves as the row vector of the data matrix. Unfortunately, merely using PCA is insufficient for a noise-free micromotion subspace, since the outliers in latent codes degrade the quality of the extracted space. As such, we further turn to a classical technique called *robust PCA* (Wright et al., 2009), which can recover the underlining low-rank space from the latent codes with sparse gross corruptions. It can be formulated as a convex minimization of a nuclear norm plus an $\ell_1$ norm and solved efficiently with alternating directions optimization (Candès et al., 2011). Through the principal component of the subspace, we get a robust micromotion edit direction $\Delta\mathbf{V}$.

**Step 3: Applying the subspace transformation.** Once the edit direction is obtained, we could edit any arbitrary input faces for the micromotion. Specifically, the editing is conducted simply through interpolation and extrapolation along this latent direction to obtain the intermediate frames. For an arbitrary input image $I_0'$, we find its latent code $\mathbf{V}_0' = E(I_0')$, and the videos can be synthesized through

$$I_t = G(\mathbf{V}_t) = G(\mathbf{V}_0 + \alpha t \Delta\mathbf{V}), \tag{1}$$

where $\alpha$ is a parameter controlling the degree of interpolation and extrapolation, $t$ corresponds to the index of the frame, and the resulting set of frames $\{I_t\}$ collectively construct the desired micromotion such as "smiling", "eyes opening". Combining these synthesized frames, we obtain a complete video corresponding to the desired micromotion.

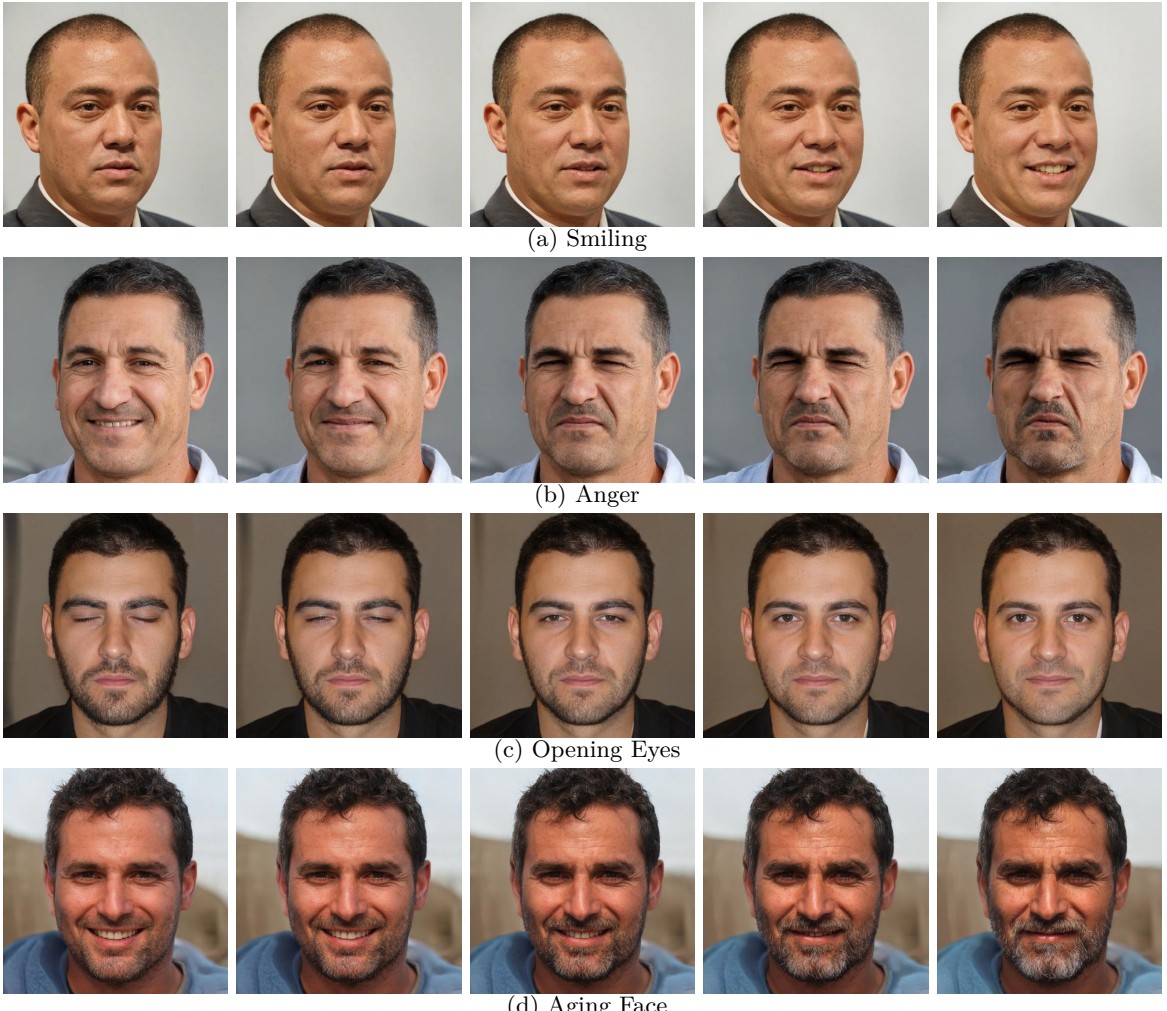

Figure 4: **Illustrations of versatile micromotions found by text-anchored method.** We decode the micromotions across different identities, and apply them to in-domain identities. From Top to Bottom: (a) Smiling (b) Anger (c) Opening Eyes (d) Aging Face. Best view when zoomed in. Please refer to supplementary for complete video sequences.

## 4    Experiments

In the experiments, we focus on the following questions related to our hypothesis and workflow:

- Could we locate subspaces for various meaningful and highly disentangled micromotions? (Sec. 4.2)
- Could we transfer decoded micromotion to other subjects in various domains? (Sec. 4.3)
- Could we extend the micromotions to novel subjects with no computation overhead? (Sec. 4.3)
- Is this framework robust towards various choices of prompts and identities? (Sec. 4.4)

In short, we want to prove two concepts in following experiments: (a) **Universality**: Our pipeline can consistently find various micromotion, and the decoded micromotion can be extended to different subjects across domains; (b) **Lightweight**: Transferring the micromotion only requires a small computation overhead.

To validate these concepts, we now analyze our framework by synthesizing micromotions. We mainly consider five micromotions as examples: (a) smiling, (b) angry, (c) opening eyes, (d) turning head, and (e) aging face. We also consider more editings when comparing with other methods. Following the workflow, we obtain the edit directions for each micromotion, and then synthesize on other cross-domain images including characters in animations, sculptures, and paintings.

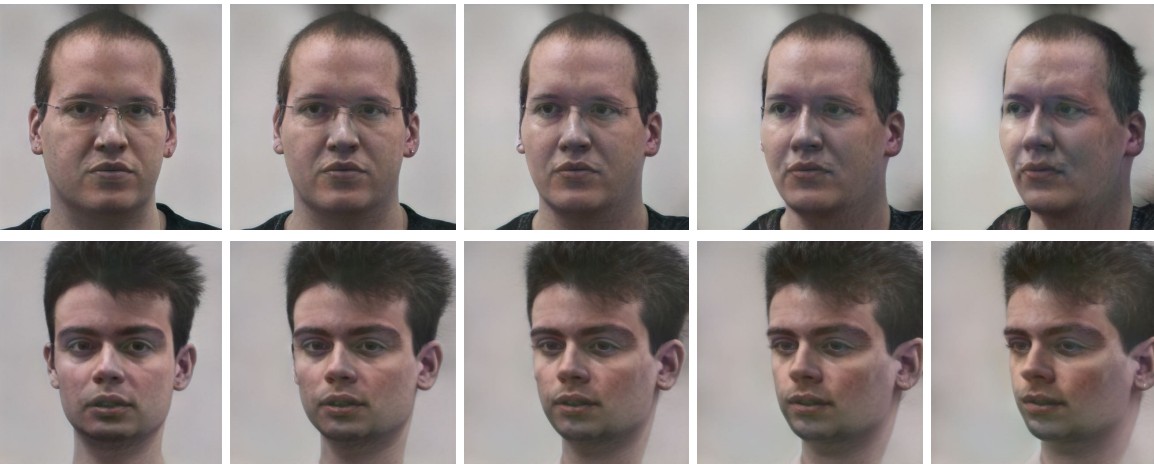

Figure 5: **Illustrations of the micromotion "turning head" found by our video-anchored method.** Best view when zoomed in. Please refer to supplementary for complete video sequences.

## 4.1 Experiment Settings

The pre-trained models (StyleGAN-v2, StyleCLIP, and StyleGAN encoders) are all loaded from the public repositories (Abdal et al., 2020; Alaluf et al., 2021b; Patashnik et al., 2021; Radford et al., 2021). When optimizing the latent codes, the learning rate is 0.1 and we use Adam optimizer. For the text-anchored and video-anchored methods, the numbers of latent codes we generate are 16 and 7. In robust PCA, 4 principal dimensions are chosen. We also search the extrapolation scale hyperparameter $\alpha$ between 0.1 and 10.

For the text-anchored experiments, the original images are generated using random latent codes in StyleGAN-v2 feature space. The text prompts we construct is in the general form of (a) "A person with {} smile"; (b) "A person with {} angry face"; (c) "A person with eyes {} closed"; (d) "{} old person with gray hair", which correspond to the micromotions of smiling, angry, eyes opening and face aging. Here, the wildcard "{}" are replaced by a combination of both qualitative adjectives set including {"no", "a big", "big", "a slight", "slight", "a large", "large", " "} and quantitative percentages set including {10%, ..., 90%, 100%}. We will discuss the choice of various text templates and their outcomes in the ablation study. For the video-anchored experiments, we consider the micromotion of turning heads. The referential frames are collected from the Pointing04 DB dataset (Gourier et al., 2004), and the frames we used for anchoring include a single identity with different postures, which has the angle of $\{-45°, -30°, -15°, 0°, 15°, 30°, 45°\}$.

## 4.2 Micromotion Subspace Decoding

In this section, we use our anchoring methods to locate the micromotion subspace, discovering the editing direction, and apply it to the in-domain identities to generate desired changes. Figure 4 and Figure 5 show the generated five micromotions using text-anchored and video-anchored methods respectively. Within each row, the five demonstrated frames are sampled from our synthesized video with the desired micromotions. As we can see, these results illustrate continuous transitions of human faces performing micromotions smoothly, which indicates the edit direction decoded from the micromotion subspace is semantically meaningful and highly disentangled. Therefore, our framework successfully locates subspaces for various micromotions.

### 4.2.1 Quantitative Analysis

To validate if our framework can consistently produce high-quality micromotions, we further compare our decoded micromotions with results from other baselines. We consider the following two methods: InterfaceGAN (Shen et al., 2020b) and GANspace (Härkönen et al., 2020). InterfaceGAN is a supervised disentangle method obtaining edit directions from trained SVMs, while GANspace is an unsupervised disentangle method that discovers edit directions from the principal components of sampled latent codes. We obtain the editing directions from these baselines respectively, performing editings on 2,000 images, and comparing results quantitatively via the following two analyses.

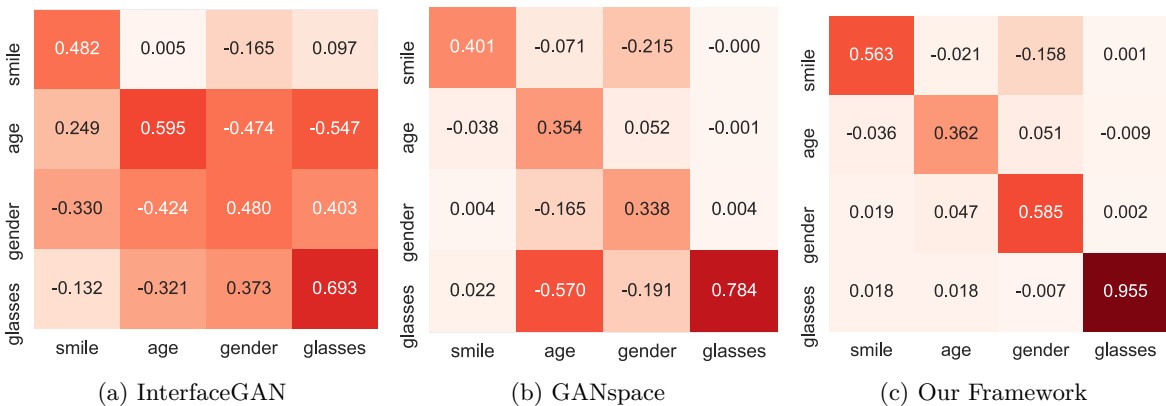

Figure 6: **Re-scoring Analysis** We conduct a re-scoring analysis similar to (Shen et al., 2020b), to quantitatively measure if our discovered editing directions can be successfully disentangled with other irrelevant attributes. We use the scores of a series of pre-trained attribute classifiers to measure how well different editing methods could depict the target attribute (e.g. smile, aging, glasses, etc.). We show a confusion matrix, where in each row we run latent-space editing with each of the attributes, and in each column shows the corresponding changes in the score by different pre-trained attribute classifiers, respectively. Higher scores indicate larger changes. We notice a clear diagonal pattern when editing latent space using our framework, indicating its good disentangle property.

**Re-scoring Analysis** We conduct a re-scoring analysis similar to (Shen et al., 2020b), to quantitatively measure if our discovered editing directions can be successfully disentangled with other irrelevant attributes. In this analysis, for a target attribute (*e.g.* smiling), we edit the input images using the editing direction discovered by our methods and baselines, and we use the scores of a series of pretrained classifiers (Na, 2021) to measure how the edits influence the target attribute as well as other non-target attributes. Ideally, a well-disentangled editing direction should result in a significant change for the target attributes, while it should have a neglectable influence on other ones.

Our results[1] can be found in Figure 6. In these three confusion matrices, for each row, we run latent-space editing with each of the attributes, and each column shows the corresponding changes measured by pretrained attribute classifiers. For example, in the matrix of InterfaceGAN, the first row means when applying the editing direction "smile" and measuring the changes using pretrained classifiers, the score of "smile" increases by 0.482 while the score of "age" also increases by 0.005, *etc.* Here, higher scores indicate larger changes. From these results, we observe that our framework produces a clear diagonal pattern, indicating a strong influence on desired attributes with little influence on non-target ones. On the other hand, the baseline methods lead to significant changes in both target and non-target attributes. Therefore, our framework demonstrates good disentangle performance compared with previous methods.

**Identity-agnostic Analysis** It is essential to preserve the identity in face editing. Following the previous work (Shen et al., 2020b), we also perform an identity-agnostic analysis. In this analysis, we use a pre-trained face identifier (Na, 2021) to quantitatively evaluate if the identity is changed after the image editing for different target attributes.

We compare our framework with other editing baselines and demonstrate the results in Table 1. Here, each row shows the changes in identity score when editing the target attributes, and a smaller score indicates better identity preservation. In this table, we

Table 1: Identity-agnostic Analysis. Each row shows the changes in identity score (the smaller the better) when editing the attributes in latent space. Here we want to make sure the attribute editing methods are identity-agnostic. We observed that our method yields the smallest changes in identity score.

|  | Smile | Age | Gender | Glass |
|---|---|---|---|---|
| InterfaceGAN | 0.0515 | 0.1294 | 0.1225 | 0.0916 |
| GANspace | 0.1081 | 0.0975 | 0.0507 | 0.1420 |
| Ours | **0.0047** | **0.0660** | **0.0491** | **0.0279** |

---

[1]Notice that InterfaceGAN does not release the classifiers they used. Therefore, we use different pretrained classifiers (Na, 2021) to conduct re-scoring analysis.

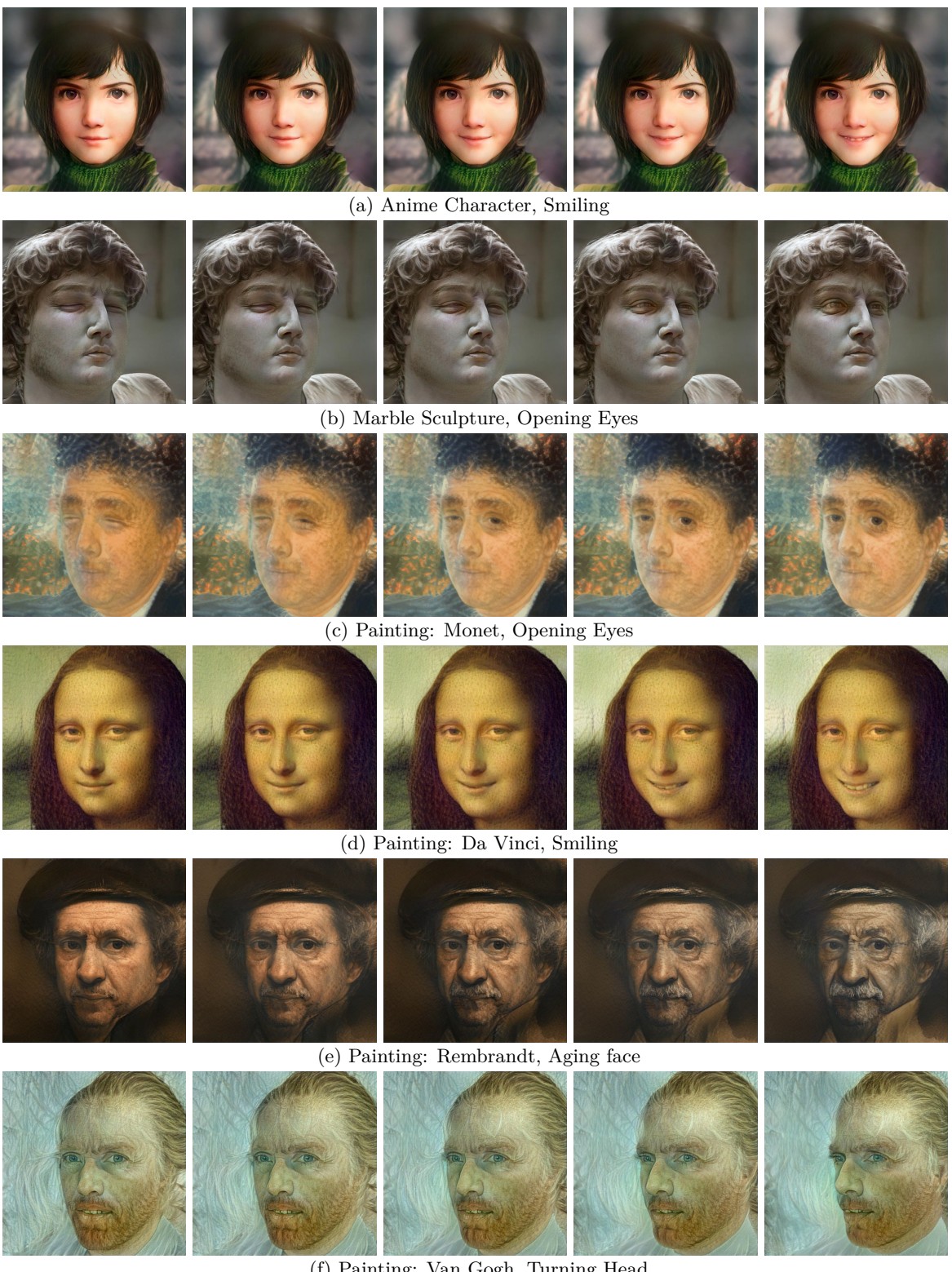

(a) Anime Character, Smiling

(b) Marble Sculpture, Opening Eyes

(c) Painting: Monet, Opening Eyes

(d) Painting: Da Vinci, Smiling

(e) Painting: Rembrandt, Aging face

(f) Painting: Van Gogh, Turning Head

Figure 7: **Micromotions on cross-domain identities.** Our micromotions generalize well when transferred to novel domains, including anime characters, sculptures, and various genres of paintings (Van Gogh, Monet, Da Vinci, Rembrandt). Best view when zoomed in. Please refer to supplementary for complete video.

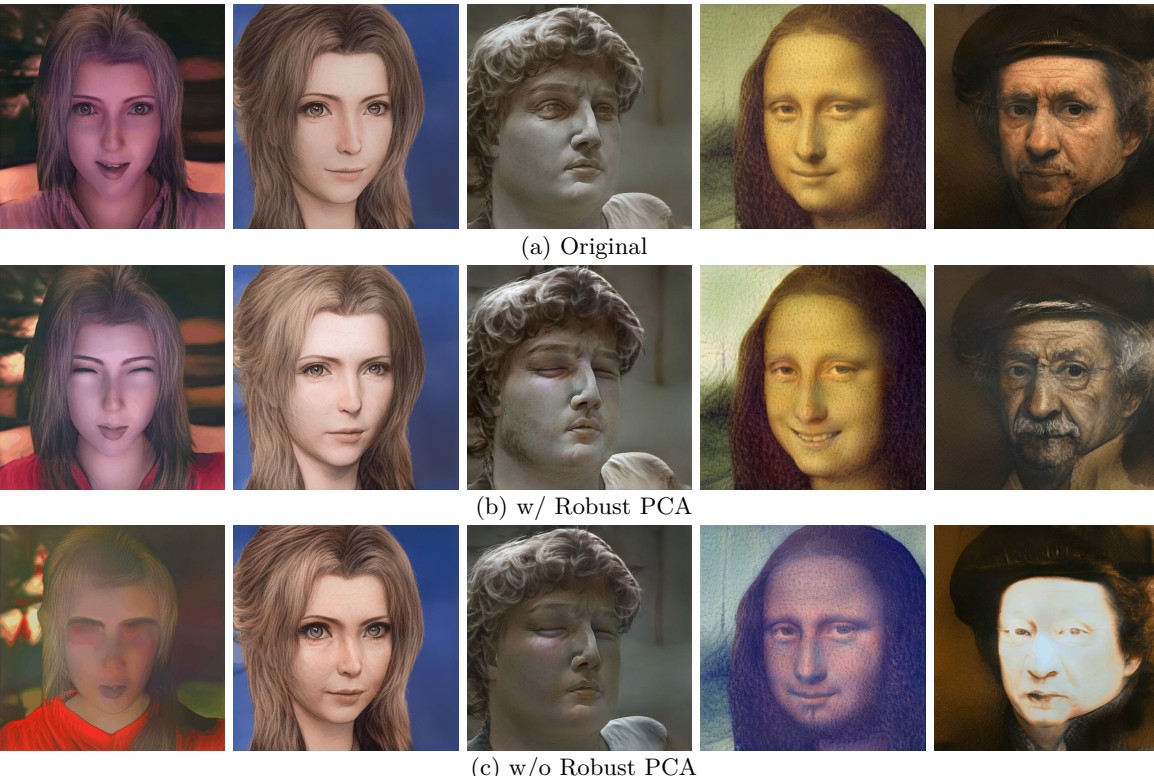

Figure 8: **Comparison between with and without Robust PCA.** For each column, from left to right, the micromotions are "closing eyes" (for the first and third columns), "angry", "smiling", and "aging face". For conciseness, we only show the original and last frame. Best view when zoomed in.

find our method preserves identities well in different edits. Other baseline methods, however, lead to more severe identity changes. Therefore, leveraging the strong latent space hypothesis, we find that our framework is able to discover more precise editing directions and better disentangles the target attributes from the human intrinsic identities.

### 4.3 Micromotion Applications on Cross-domain Identities

Sec. 4.2 decodes the micromotion from low-dimensional micromotion subspace, which verifies the first part of the hypothesis. In this section, we further verify the second part of the hypothesis, exploring if the decoded micromotion can be applied to arbitrary and cross-domain identities.

Figure 7 shows the result of transferring the decoded micromotions on novel identities. Within each row, we exert the decoded micromotions on the novel identities, synthesize the desired movements, and demonstrate sampled frames from the generated continuous videos. From these results, we observe that the sampled frames on each new identity also depict the continuous transitions of desired micromotions. This verifies that the decoded micromotions extracted from our workflow can be successfully transited to the out-domain identities, generating smooth and natural transformations. Also, this shows the low-dimensional micromotion subspace in StyleGAN is indeed not isolated nor tied to certain identities. On the contrary, leveraging the low-rank micromotion hypothesis in StyleGAN latent space, the *identity-agnostic* micromotions can be found using our framework and can be ubiquitously applied to those even out-of-domain identities.

Moreover, we emphasize that to generate dynamic micromotion on a novel identity, the entire computational cost boils down to inverting the identity into latent space and then extrapolating along the obtained edit direction, without the requirement of retraining the model or conducting identity-specific computations. This leads to effortless editing of new identity images using the found direction, with little extra cost.

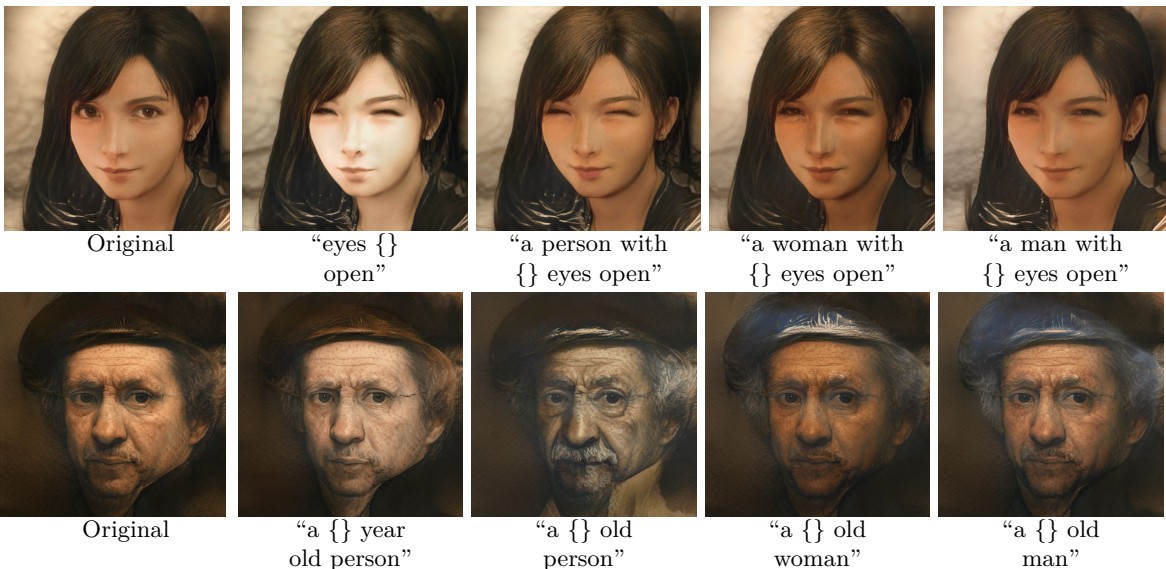

| Original | "eyes {} open" | "a person with {} eyes open" | "a woman with {} eyes open" | "a man with {} eyes open" |

| Original | "a {} year old person" | "a {} old person" | "a {} old woman" | "a {} old man" |

Figure 9: **Ablation on the choice of text template for micromotion "opening eyes" and "aging face".** For each template, we fill the wildcard "{}" using descriptive text, including {10%, 20%, ..., 100%}, {10, 20, ..., 60}, and {small, big, ...}. For conciseness, we only show the last frame of each group. Best view when zoomed in.

### 4.4 Design Choices and Ablation Study

**Ablation on component decomposition in micromotion subspace**  To verify the effectiveness of the robust decomposition in our workflow, instead of doing robust PCA to decompose the low-rank micromotion space, we sample two anchoring latent codes and adopt its interpolated linear space as the low-rank space. Then, we compare the qualitative results of the decoded micromotions. Results in Figure 8 show that synthesized videos without robust space decomposition step incur many undesired artifacts, often entangling many noisy attributes not belonging to the original and presumably mixed from other identities. Adding a robustness aware subspace decomposition, however, effectively extracts more stable and clearly disentangled linear subspace dimensions in the presence of feature fluctuations and outliers.

**Ablation on text templates**  To explore the sensitivity of the micromotion subspace w.r.t the text templates, we study various text templates that describe the same micromotion. In Figure 9 top row, we can see that the micromotion "closing eyes" is agnostic to the choice of different text templates and generate similar visual results. On the other hand, In Figure 9 bottom row, we observe the opposite where the micromotion "face aging" is sensitive to different text templates, which results in diverse visual patterns. This suggests the choice of text template may influence the performance of some micromotions, and a high-quality text guidance based on prompts engineering or prompts learning could be interesting future work.

**Ablations on number of anchors and identities**  In our framework, we rely on a series of latent codes to anchor the low-rank space, and these codes are obtained from one identity performing micromotion. Therefore, we further ask two questions for our design: (a) How would the number of latent codes influence the editing performance? (b) Could we obtain better editing results from multiple identities? For the first question, we hypothesize the number of latent codes would influence the quality of the discovered low-dimensional space, therefore influencing the editing performance. For the second question, although we have obtained high-quality micromotion editing direction from a single identity, we explore if multiple identities decrease the correlation between micromotion and identity and lead to better disentanglement.

We use different numbers of anchoring latent codes and identities to discover the editing direction and apply it to novel images. In the first study, multiple latent codes are used to determine the low-rank space. Notice that when using only one anchoring latent code, the framework reduces to using StyleCLIP to find editing

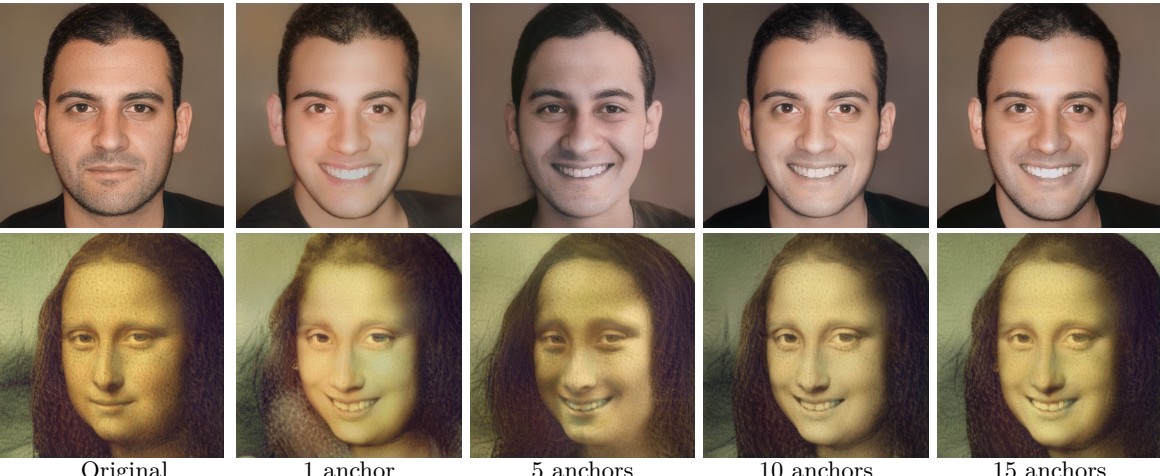

Figure 10: **Ablation on the number of anchoring latent codes used to find the low-rank latent space.** We use the micromotion "smiling" as an example, while applying to both in-domain (human face) and out-of-domain (painting) images, we notice the quality of latent space editing improves proportionally w.r.t the number of anchors until it finally saturated at around 10 anchors.

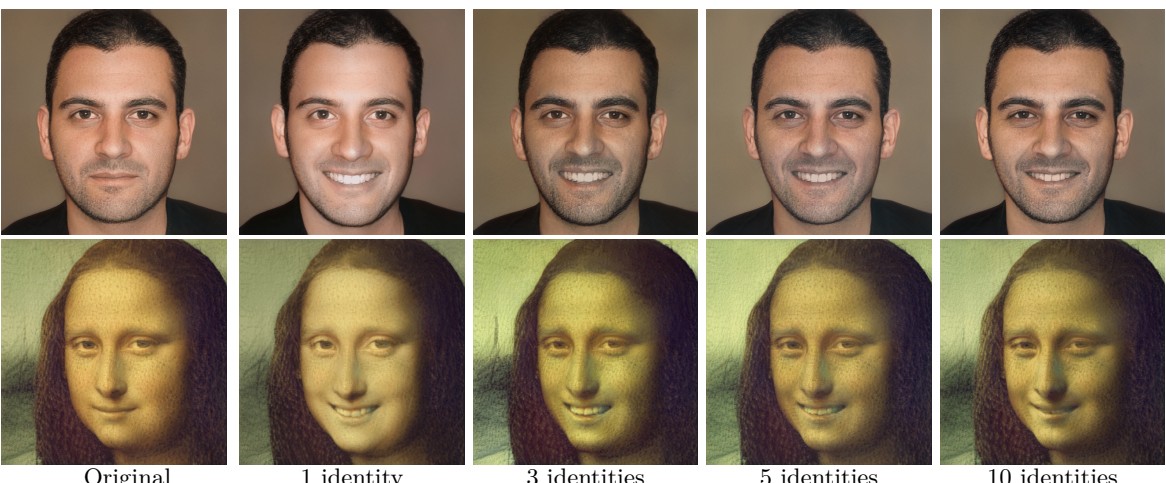

Figure 11: **Ablation on the number of identities used in latent-space optimization.** We use the micromotion "smiling" as an example while applying to both in-domain (human face) and out-of-domain (painting) images. We notice there is no clear visual improvement in the quality of micromotion as the number of identities grows, however it would result in different styles of "smile".

direction and directly apply it to novel images. In the second study, with multiple identities, we optimize the latent code on each identity separately and use the average latent code as the final editing direction.

Our results can be found in Fig 10 and Fig 11. For the effects of anchoring latent codes, we observe that fewer anchors result in noises and artifacts, indicating insufficient disentanglement. Meanwhile, we observe the quality of latent space editing improves gradually with respect to the number of anchors. This motivates us to use a series of anchoring latent codes for a better low-rank latent space. For the effects of identities, we observe that although using more identities has some weak benefits (*e.g.* better background preservation), there is no clear visual improvement compared with using one identity. This motivates us to stick with one identity with better efficiency in our framework.

## 5 Conclusions

In this work, we analyze the latent space of StyleGAN-v2, demonstrating that although trained with static images, the StyleGAN still captures temporal micromotion representation in its feature space. We find versatile micromotions can be represented by low-dimensional subspaces of the original StyleGAN latent space, and such representations are disentangled and agnostic to the choice of identities. Based on this finding, we explore and successfully decode representative micromotion subspace by two methods: text-anchored and video-anchored reference generation, and these micromotions can be applied to arbitrary cross-domain subjects, even for the virtual figures including oil paintings, sculptures, and anime characters. We also explore the various design choices and their corresponding effects on our framework. Future works may study more complex motion subspace and further explore if latent space corresponding to larger-scale motion is also ubiquitous and can be disentangled from other attributes and identities, which could potentially lead to many interesting applications.

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

# A    Ablation on the identities of anchoring latent codes

In this ablation, we explore if the choice of identity influences the micromotion quality. We use photos of different people (denoted as Identity A, B, and C) to discover editing directions, and we generalize to the same sketch painting. The result is shown in Figure 12. We observe that latent codes decoded from various identities generate visually similar micromotions. Therefore, the micromotion can be decoded using different identities and still result in semantically correct edits.

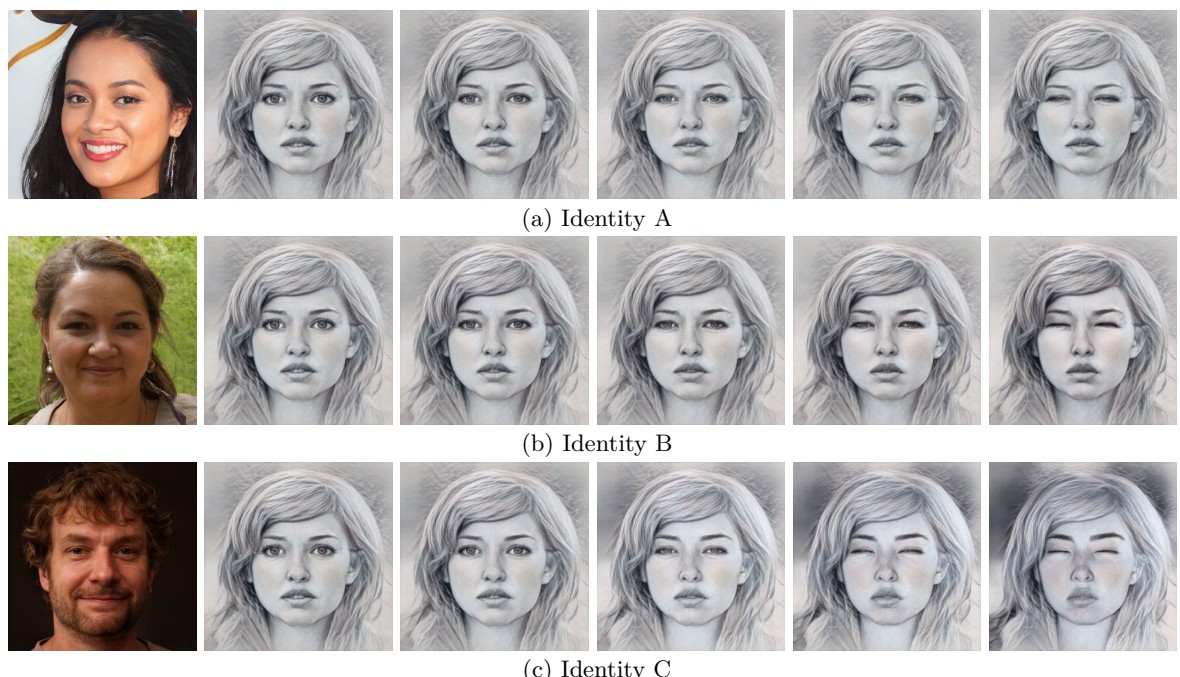

(a) Identity A

(b) Identity B

(c) Identity C

Figure 12: **Ablations on the identities of anchoring latent codes.** The first column shows synthesized images of three different identities generated by three latent codes. From the second column, we show the micromotion subspace (i.e. "closing eyes") decoded from three identities in the first column, exhibiting visually similar results, when generalized to a sketch painting in the novel domain.

# B    Additional examples of micromotions transferred to novel domains

In Figure 13, we include additional visual examples to demonstrate that our micromotions generalize well when transferred to novel domains. The additional novel domains include bronze sculptures, oil/sketch painting, and more anime characters.

# C    Supplementary Video

We present a series of in-domain and out-of-domain synthesized videos to fully demonstrate the visual effects of our method. Those videos can be found in the "video" folder of the supplementary materials.

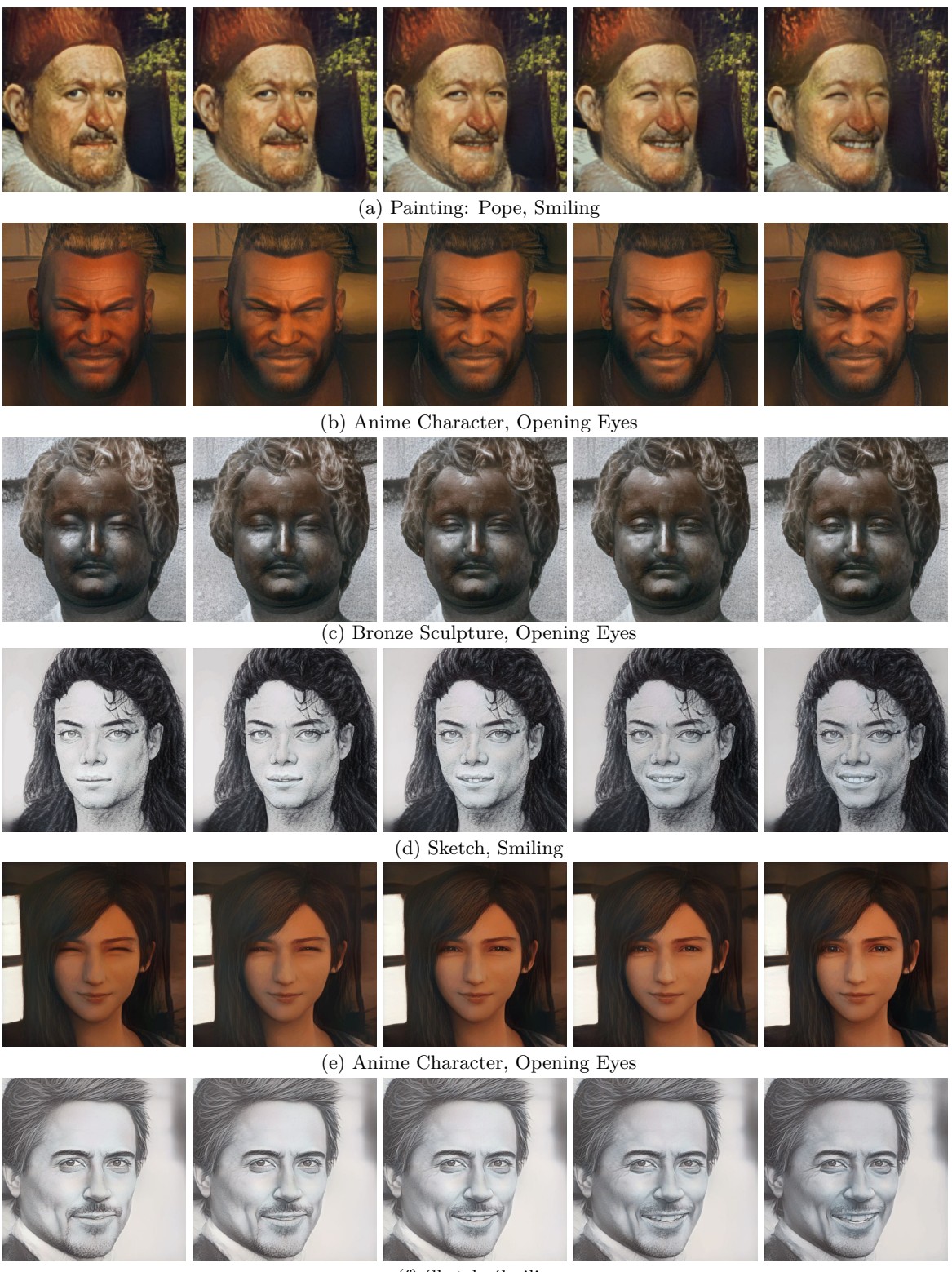

(a) Painting: Pope, Smiling

(b) Anime Character, Opening Eyes

(c) Bronze Sculpture, Opening Eyes

(d) Sketch, Smiling

(e) Anime Character, Opening Eyes

(f) Sketch, Smiling

Figure 13: **Additional examples of micromotions transferred to novel domains.** Our micromotions generalize well when transferred to novel domains, which include anime characters, sculptures, and various genres of paintings (oil painting, sketch). Best view when zoomed in. Please refer to supplementary for complete video sequences.

## D  Qualitative comparison with baselines

In this experiment, we qualitatively compare the performance of our method with existing editing methods. We consider the following baselines: InterfaceGAN Shen et al. (2020a), GANspace Härkönen et al. (2020), StyleFlow Abdal et al. (2021b), Zhuang et al. (2021), and MoCoGAN-HD Tian et al. (2021).

**Experiment Setting.**    All methods are tested on the StyleGAN-v2 pretrained on the FFHQ dataset, and we follow the settings stated in Sec. 4.1. We adopt the released pretrained models for most of the baselines except Zhuang et al. and StyleFlow. For Zhuang et al., since it is only trained on $256 \times 256$ images, we first attempt to train the model on $1024 \times 1024$ using the code released by authors for a fair comparison. However, the model does not converge on 1024 resolution. Therefore, we perform comparisons by collecting the source and edited images shown on their papers and using our method to perform the edit. For StyleFlow, since the code is in Tensorflow 1.x and not supported by our machines, we also compare with the images on their papers qualitatively. For most baselines, we compare their editing results on two representative target attributes in the paper: "smiling" and "aging". We choose these two attributes because these are the common attributes explored by both our method and baselines. Finally, for MoCoGAN-HD, since it is only trained on the "talking-head" task on FFHQ dataset, we compare the editing performance on this task. Specifically, we apply our video-anchored method to synthesize videos of a person talking using a single input reference video. Meanwhile, videos of MoCoGAN-HD are synthesized using their pretrained models.

**Results.**    We first demonstrate the comparison between our method and InterfaceGAN, GANspace. The comparison result is shown in Figure 14. From the result, we observe that our method demonstrates comparable or better performance than existing baselines. Specifically, compared with these methods, our method faithfully preserves the identities of the edited subjects, while InterfaceGAN often changes the identities, even genders, during edition, and GANspace usually produces minor edit towards target attributes. Also, to see the benefits of on-demand disentanglement vs. manually picking from principal components, we demonstrate the first 14 principal directions from GANspace in Figure 17. We observe that (1) all of these latent codes cannot open/close people's eyes, meaning GANspace fails to find editing directions for this attribute even after manually checking 14 directions in this case; (2) some of the editing directions are clustered, e.g, attribute "glasses" is entangled with "age" in C3, and is entangled with "gender" in C9. On the other hand, our method can find editing directions for these attributes without heuristic human choices.

Besides, we demonstrate the comparison between our method and MoCoGAN-HD in Figure 15. In the figure, the first six rows show synthesized videos by our methods. Starting from a single input video ($1^{st}$ and $4^{th}$ row), our video-anchored method inverts their frames, reconstructs the original video ($2^{nd}$ and $5^{th}$ row) and produces the same talking actions on a new identity ($3^{rd}$ and $6^{th}$ row). The last two rows show the result of MoCoGAN-HD. From the results, we highlight two benefits of our methods: First, videos synthesized by our method has more significant variance than MoCoGAN-HD. Our synthesized videos show a person talking with mouth and eyes actions, while most frames in MoCoGAN-HD resemble the first frame and have little changes. Second, our method allows on-demand talking actions, *i.e.*, the synthesized video resembles the reference video at each frame. On the other hand, MoCoGAN-HD cannot control the synthesized talking action. Besides these two benefits, we emphasize that our method only requires a single input video to generate talking motions on novel identity, while MoCoGAN-HD requires a large training dataset (*e.g.,* VoxCeleb Nagrani et al. (2017) dataset with 22,496 clips). With these advantages, we conclude that our method is more effective and convenient than MoCoGAN-HD in the talking-head task.

Finally, the comparison between our method and StyleFlow, Zhuang et al. are shown in Figure 16[2]. We observe that our method and two baseline methods result in different styles of "smile" and "aging", while the quality is comparable. Besides, we emphasize that both StyleFlow and Zhuang et al. require training auxiliary models, while our method does not need any new models, and finding editing direction using our method can be done in a few minutes. Therefore, our method outperforms the listed baselines by providing a more convenient editing framework.

---

[2]The original images used in our work are inverted from the corresponding input images for baselines. Therefore, the original images are slightly different.

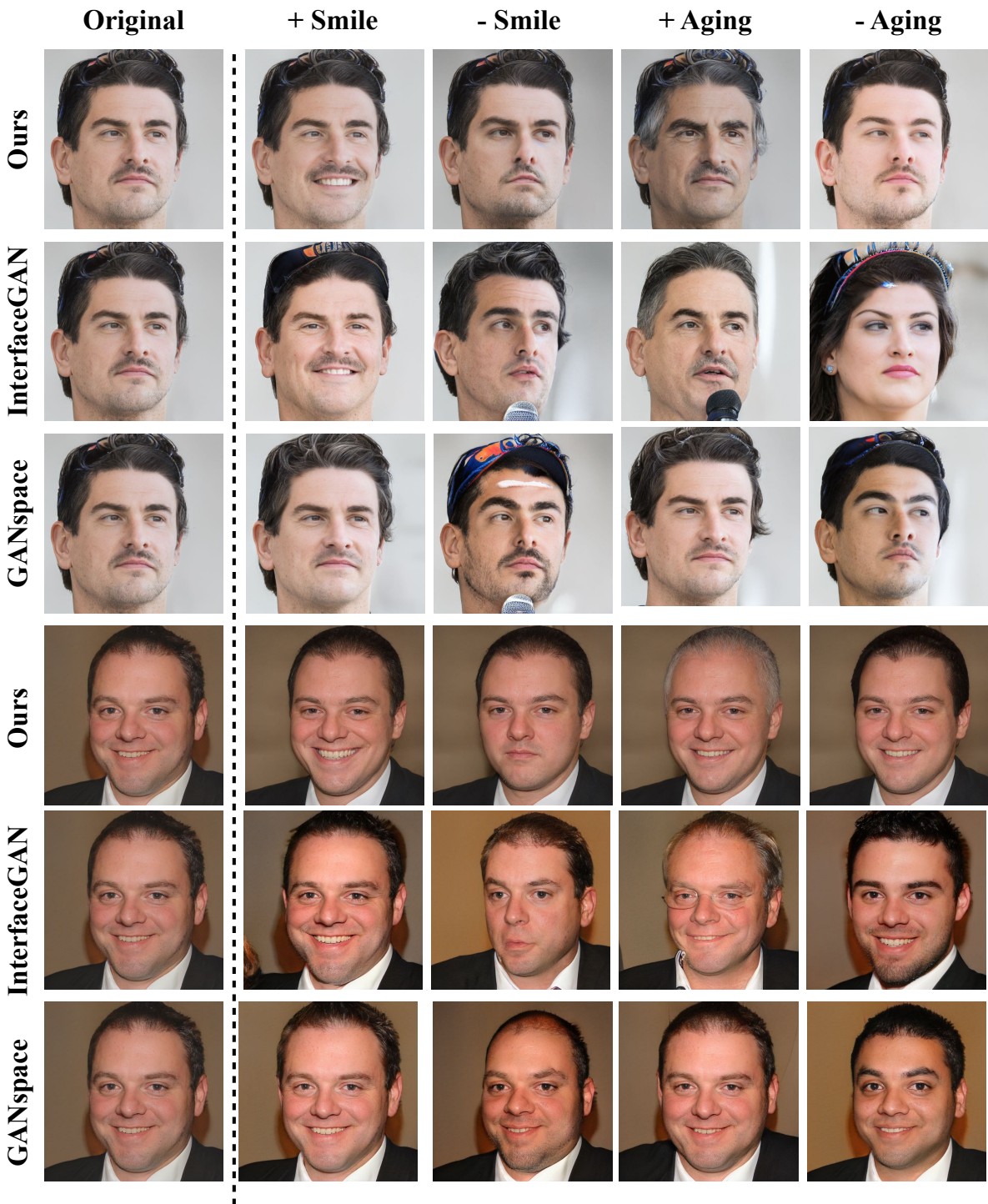

Figure 14: **Qualitative comparison with InterfaceGAN and GANspace.** The target attributes are "smiling", "aging" for human faces.

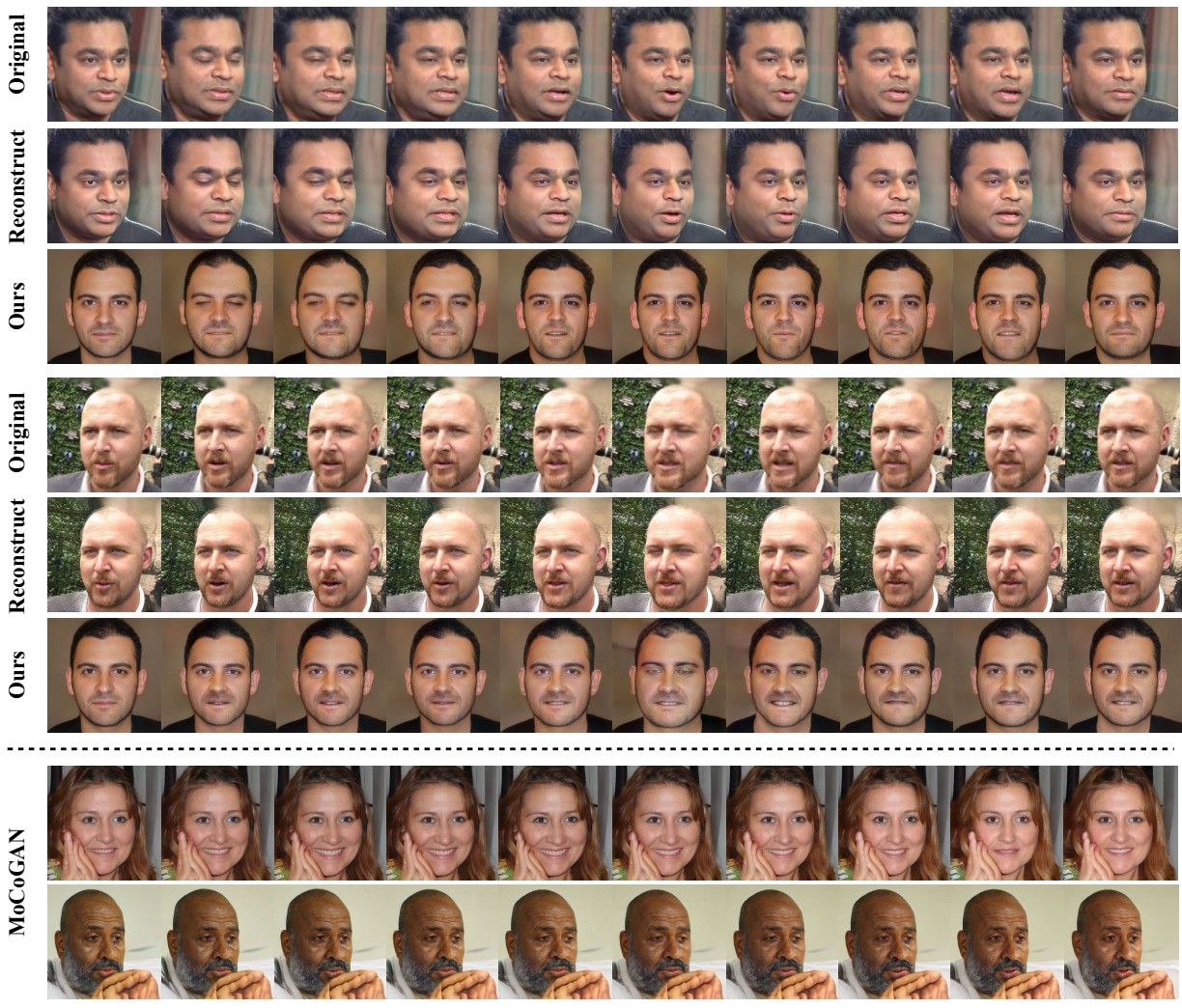

Figure 15: **Qualitative comparison with MoCoGAN-HD on task "talking head".** The first six rows show synthesized videos by our methods. Starting from single input video (1st and 4th row), our video-anchored method invert their frames, reconstruct the original video (2rd and 5th row) and produce the same talking actions on new identity (3rd and 6th row). The last two rows show the result of MoCoGAN-HD.

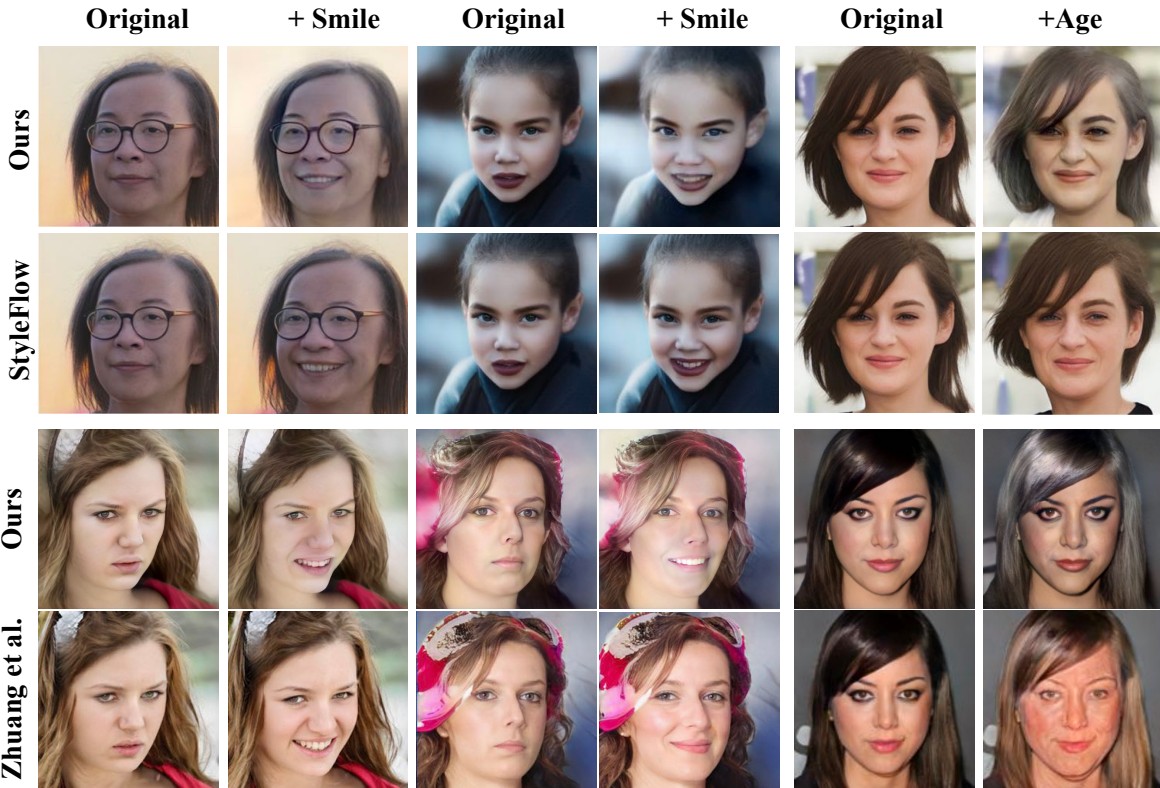

Figure 16: **Qualitative comparison with StyleFlow and Zhuang et al.**

# E  Ablation on subspace decomposition techniques

We conducted a further ablation study, by comparing different subspace decomposition techniques, including Robust PCA, Vanilla PCA, and without PCA. In Figure 18, we show Robust PCA yields the best visual results, followed by Vanilla PCA, while without PCA yields results with the worst visual quality. When comparing the results using vanilla PCA with robust PCA, we can observe the former creates more undesired artifacts. For example, in the third column of Figure 18, we observe vanilla PCA create an unwanted artifact around the shoulder of the sculpture, while robust PCA provides a cleaner image. On the other hand, micromotion subspace without PCA decomposition creates images with the worst quality. Most of them have serve distortion and the faces are barely recognizable. The ablation demonstrates vanilla PCA is insufficient for a noise-free micromotion subspace, while the Robust PCA is a more favorable choice.

# F  Ablation on different GANs

Besides the StyleGANv2 discussed in main paper, in this ablation, we further discuss if we can disentangle from progressive GAN Karras et al. (2018) and BigGAN Brock et al. (2018).

**Experiment Setting.**  For both progressive GAN and BigGAN, we adopt the publicly released pretrained models. The progressiveGAN is loaded from pytorch hub. The BigGAN is loaded from the original repository. For the editing tasks, we choose the target attributes according to their training datasets. Specifically, for progressive GAN, the model is trained on CelebA Liu et al. (2015) dataset, and we study the attributes "smiling" and "aging" on human faces. For BigGAN, the model is trained on ImageNet Deng et al. (2009) dataset, we study the attributes "opening mouth" and "closing eyes" on dogs. We use the same text prompts for the human face experiment ("A person with {} smile", "{} old person with gray hair"). For the experiment on dogs, the text prompts we construct are in a similar form ("A dog with eyes {} close", "A dog with mouth {} open").

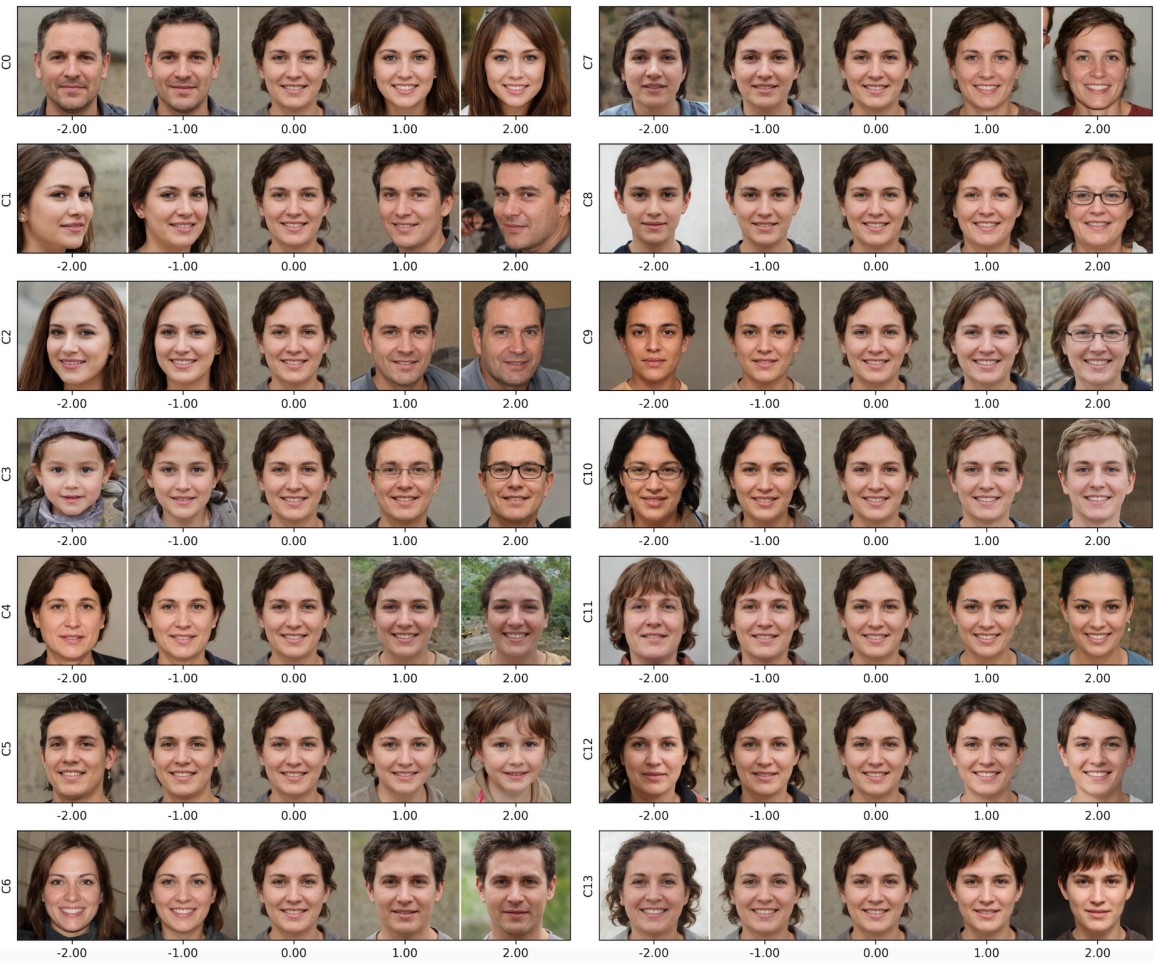

Figure 17: **Principal Components of GANspace.** We demonstrate the editing effects of using the first 14 principal components to edit images.

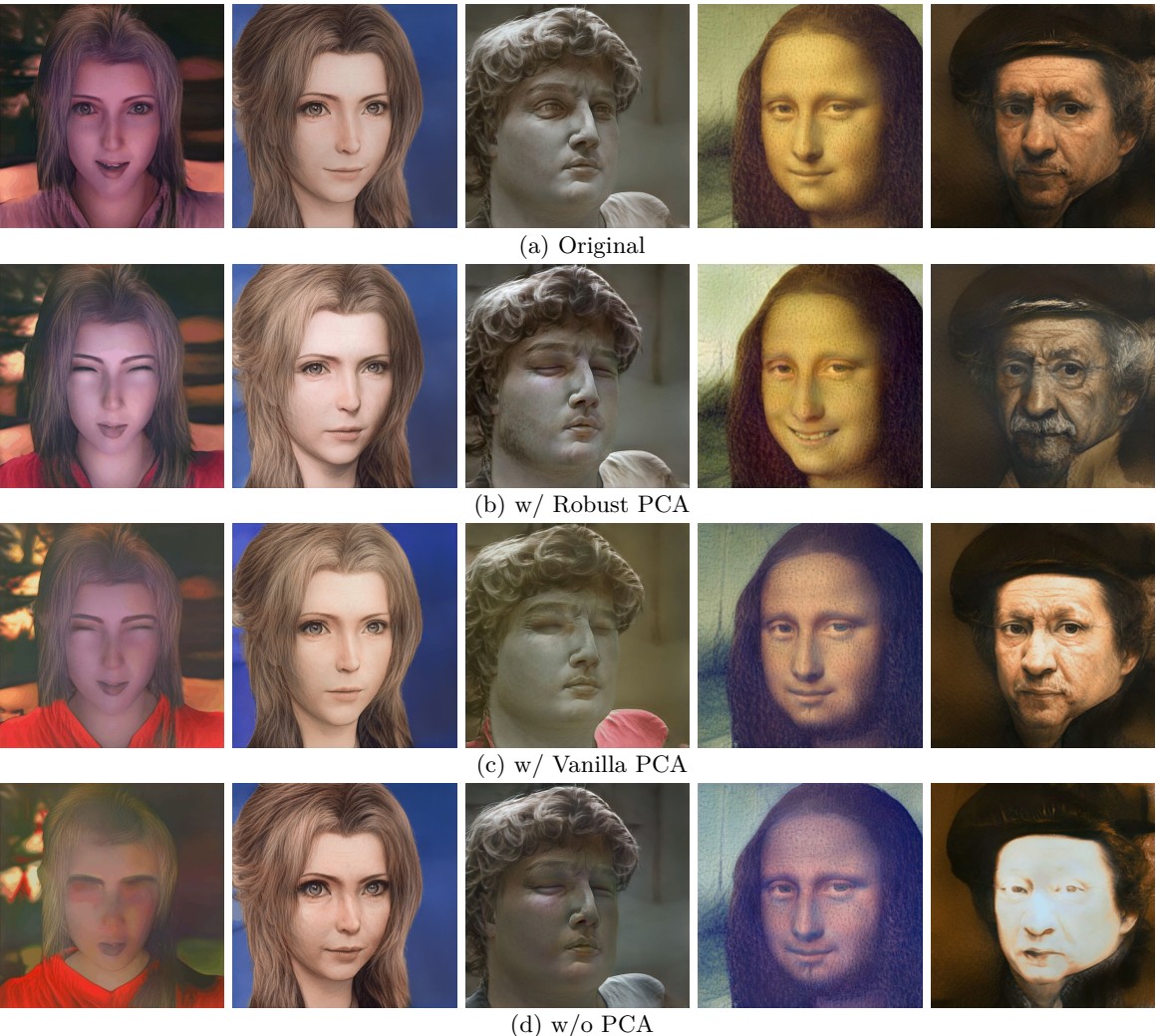

(a) Original

(b) w/ Robust PCA

(c) w/ Vanilla PCA

(d) w/o PCA

Figure 18: **Comparison between Vanilla PCA, Robust PCA and without PCA.** For each column, from left to right, the micromotions are "closing eyes" (for the first and third columns), "angry", "smiling", and "aging face". For conciseness, we only show the original and last frame. Best view when zoomed in.

**Results.** The result is shown in Figure 19. From the figure, we observe that both GANs do not synthesize high-quality editing images. For example, for the "opening mouth" attribute in BigGAN, the mouths of dogs in the first two rows are larger, but both the dogs and backgrounds change drastically. This is even worse for the target attribute "closing eyes". Similarly, in Progressive GAN, we find slight changes toward target attributes "smiling" and "aging", while the identities are largely changed. This result indicates the latent space in BigGAN and progressive GAN are not highly disentangled. There are two possible reasons: First, the latent code dimension in BigGAN and Progressive GAN ($1 \times 512$) is smaller than the one in StyleGANv2 ($18 \times 512$). Second, the hierarchical structure in StyleGAN might lead to better disentanglement. Therefore, compared with these generators, the StyleGANv2 used in the main paper is a better choice.

## G   Ablation on inversion methods

In this experiment, we demonstrate how different inversion methods influence editing performance. We consider three methods: Restyle+pSp Alaluf et al. (2021b); Richardson et al. (2021), Restyle+e4e Tov et al. (2021), and vanilla e4e method. The result is demonstrated in Figure 20. We observe that using different

| **Original** | **Closing Eyes** | **Opening Mouth** | **Original** | **Smiling** | **Aging** |
|---|---|---|---|---|---|

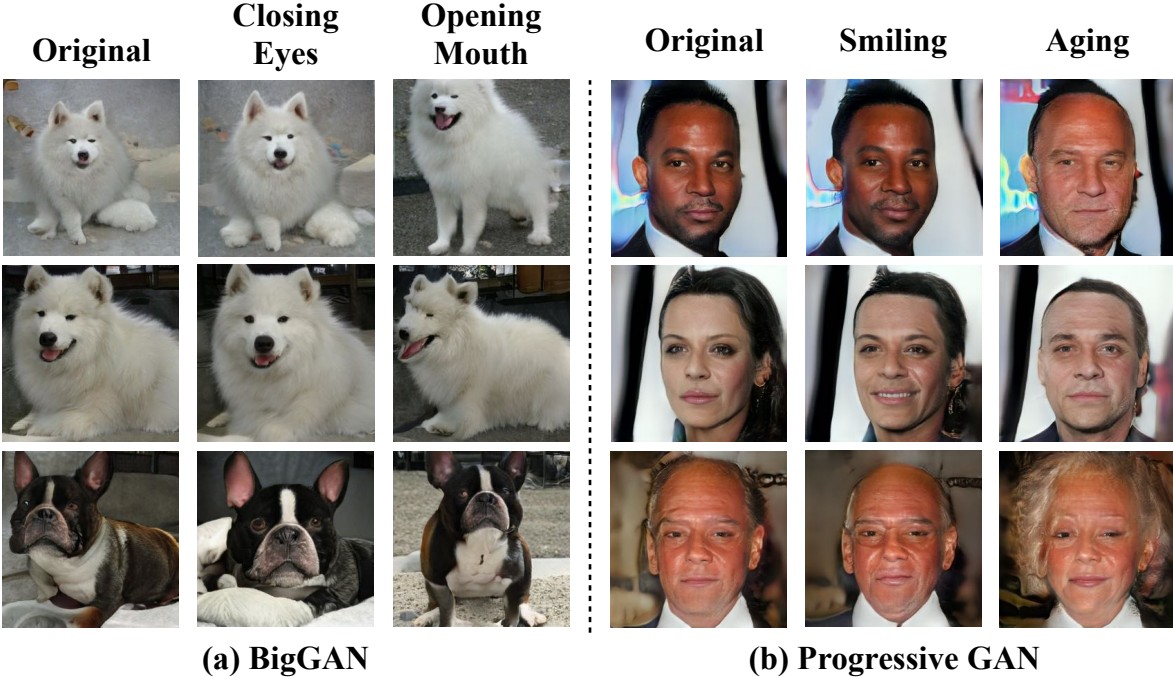

**(a) BigGAN**        **(b) Progressive GAN**

Figure 19: **Ablation on different GANs.** We demonstrate the editing results using different GANs, including BigGAN (left) and Progressive GAN (right). The target attributes are "closing eyes", "opening mouth" for dogs, and "smiling", "aging" for human faces.

inversion methods influences the editing results. Arguably, Restyle+pSp preserves the background color and details best. Besides, we also observe that other methods produce undesired changes (*e.g.,* images are darker for the upper rows). Choosing a faithful inversion method helps produce high-quality edits.

## H    Failure case

While our method can disentangle many micromotions and transfer to novel images in different fields, we would like to demonstrate a few limitations of our framework. First, the editing ability of our method originates from the disentangled latent space of StyleGANv2. We have shown in section F that, with a less disentangled GAN architecture, this framework cannot produce high-quality editing results. Second, when transferring the editing directions to out-of-domain images, we first need to invert the input images to vectors in the latent space of StyleGAN. When the input largely deviates from a photo-realistic person, the inversion model fails to find the corresponding latent code, and therefore the editing will also fail. We provide an example in Figure 21. Here, the target attribute is "smiling", and the input images are anime characters (Mario, Chihiro). We find that the latent code produced by the encoder cannot reconstruct the images (in the second and fifth row), and therefore the editing images have poor quality.

## I    Broader impact

We acknowledge that facial editing and video synthesis techniques can be harmful if misused. For example, they can be used to forge fake videos containing offenses and misinformation. However, our method does not try to prompt these societal consequences, but to develop a method that conveniently produces motions for common use. To prevent misuse, our method can be combined with the idea in Yu et al. (2020) by adding distinct fingerprints for synthesized videos.

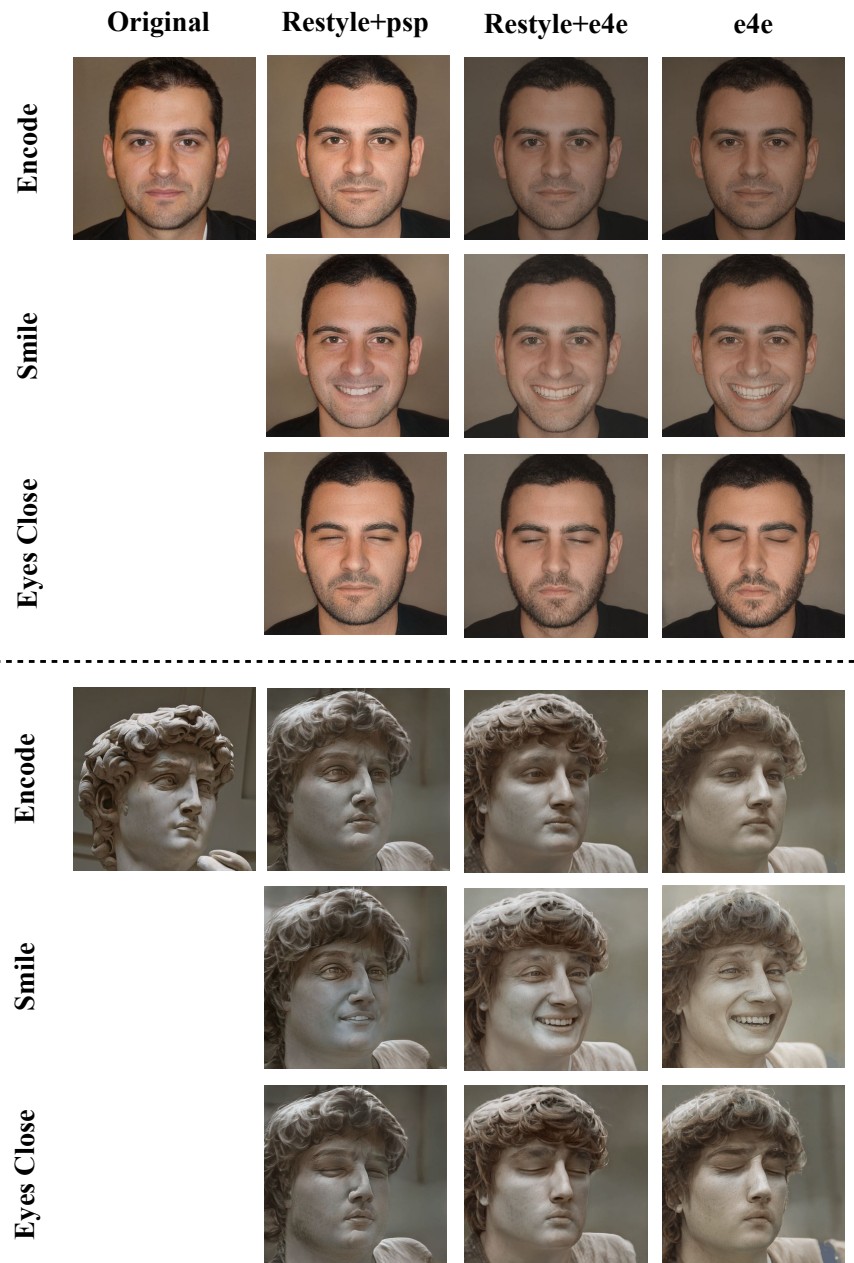

Figure 20: **Ablation on inversion methods.** We demonstrate editing results of using different inversion methods for both in-domain and out-of-domain input image.

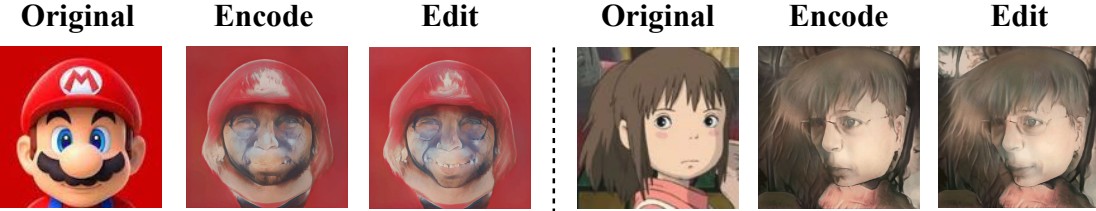

Figure 21: **Failure case study.** The target attribute is "smiling". We demonstrate that when the encoder fails to encode out-of-domain images (*e.g., Mario, Chihiro*), using the discovered editing direction will also synthesize incorrect image.

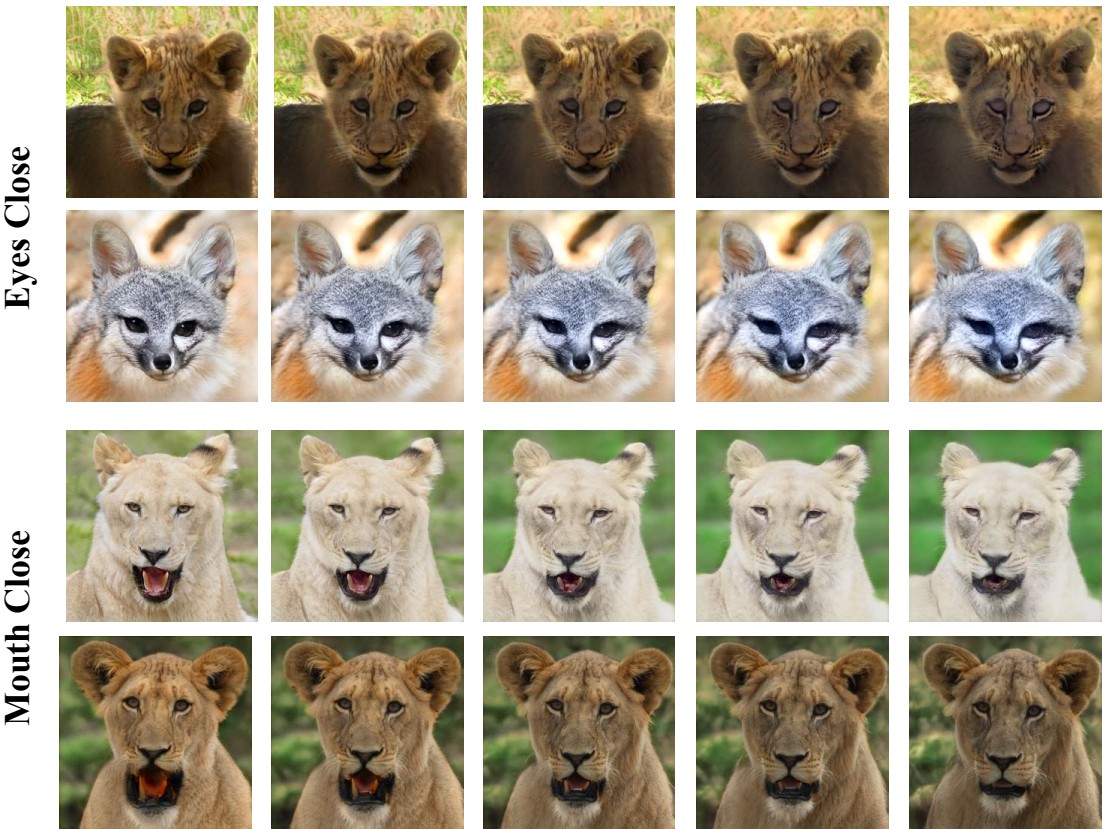

Figure 22: **Micromotions on wild animals.** The target micromotions are "eyes close" and "mouth close".

## J    Micromotions other than human face

Finally, we provide more micromotion examples on subjects other than the human face. In this experiment, we explore the micromotions of wild animals. The StyleGANv2 model we used is pretrained on the AFHQ-wild dataset Choi et al. (2020) with $512 \times 512$ resolution, and we consider "eyes close" and "mouth close" as two representative micromotion examples. The results can be found in Figure 22. From the figure, we observe that our method can also synthesize micromotions on wild animals, while the quality is not as good as those on human faces. We highlight two drawbacks here. First, the synthesized images change the background as well. Second, the synthesized images sometimes do not reflect a smooth micromotion. We provide one example in the first row. Specifically, we expect the wild animal to gradually close its eyes, while the synthesized images demonstrate a pixel-wise interpolation from open eyes to close eyes. We hypothesize this is due to the AFHQ-wild dataset does not contain wild animals with different eyes open degrees. As such, interpolation on the editing direction cannot synthesize animals with eyes half-open, which is hardly seen in the training dataset. We believe that with a high-quality dataset and better-pretrained generator, we can expect better micromotions.

