# OpenReview forum: "Grasping the Arrow of Time from the Singularity: Decoding Micromotion in Low-dimensional Latent Spaces from StyleGAN"
_TMLR — Rejected by TMLR_

### Review · Reviewer_3YL2 · 2022-11-16

**Summary Of Contributions:**

This paper aims to address the StyleGANv2 based image editing via latent editing. The authors propose a simple low-rank method to disentangle the micromotion like expressions/emotions, heads movements, and aging effects. The main idea is to find a latent path through reference anchor and GAN inversion technic, and then edit latent code via such latent path.

The main workflow contains three steps. 1) Obtain the reference anchoring via either text or video frames. 2) decompose the latent space via robust PCA for approximated micromotion subspace. 3) apply the micromotion subspace to novel identities.
It is claimed that the proposed method is simple and efficient for learning universal and highly disentangled editing directions. However, the critical baseline comparisons and analyses are missing.

**Audience:**

Yes

**Broader Impact Concerns:**

The face editing technic may have deepfake issues. A Broader Impact Statement is required.

**Claims And Evidence:**

Yes

**Requested Changes:**

1) Adding qualitative comparison with [1]
2) Adding comparison with [2]
3) Adding ablation study of GAN inversion methods.
4) Show results with more complex micromotions.
5) Add failure cases and limitation discussion.


[1] Yujun Shen, Ceyuan Yang, Xiaoou Tang, and Bolei Zhou. Interfacegan: Interpreting the disentangled face representation learned by gans, CVPR 2020.

[2] Peiye Zhuang, Oluwasanmi Koyejo, Alexander G. Schwing, Enjoy your editing: controllable GANs for Image Editing via Latent Space Navigation, ICLR 2021.

**Strengths And Weaknesses:**

Strengths:

1) This paper introduces a simple method via reference anchor and robust PCA to achieve universal and disentangled latent editing.
2) The anchor format allows either text or video frames, which is flexible for practical application.
3) The idea is intuitive and the paper is easy to follow.

Weaknesses:

1) My main concern is the lack of important baseline comparisons. Current experiments can not support the claims well.
- The introduced method is closely related to [1]. The goal of InterfaceGAN [1] is to find the latent path through a classifier trained with annotated data. The proposed method chooses a different way to find the latent path, I.e., using robust PCA with provided reference anchor. While the usage of PCA for subspace decomposition is straightforward. More importantly, the qualitative comparison with [1] is missing.
- Another closely related work [2] is also not compared.  [2] introduces a way to disentangle latent paths and a quantitative metric to evaluate latent path disentanglement. Besides, [2] also discusses the difference between universal and local directions on several GAN models. The universal directions may not work on GAN models like progressiveGAN. It is questionable if the hypothesis in Fig. 2 can also be generalized to other GAN models. More discussion would be helpful.

2) The ablation and discussion on inversion methods are missing. The proposed method relies on GAN inversion methods to find the latent code. However, different inversion methods may affect the performance significantly.

3) The illustrated micromotion is limited. The idea of finding paths via reference is useful when attributes are hard to define via text. For example, the proposed method can use real video to drive images.  While in the results, only results of simple expressions (e.g., smile and opening eyes) are provided which have also been demonstrated in [1]. More complex expressions would be good examples.

4) The failure cases and limitations of the proposed method are not provided. It is important to analyze the working scope of the proposed approach to benefit the community.

[1] Yujun Shen, Ceyuan Yang, Xiaoou Tang, and Bolei Zhou. Interfacegan: Interpreting the disentangled face representation learned by gans, CVPR 2020.

[2] Peiye Zhuang, Oluwasanmi Koyejo, Alexander G. Schwing, Enjoy your editing: controllable GANs for Image Editing via Latent Space Navigation, ICLR 2021.

---

> ### Author Response · Authors · 2022-12-03
> **Author Response to Reviewer 3YL2 (1/2)**
>
> We thank reviewer 3YL2 for all the comments.
>
> **Q**: My main concern is the lack of important baseline comparisons. Current experiments can not support the claims well.
> **A**: We thank reviewer 3YL2 for suggesting more baseline comparisons. In the paper, we compared our method with InterfaceGAN and GANspace quantitatively to demonstrate the disentanglement ability of our method (Sec. 4.2.1). Here, we further include qualitative comparisons with these methods, and these results are in the updated PDF (Sec. D). Besides, we also compare with more baselines in Figure 14-17. Generally speaking, our method produces better or comparable images with much simpler framework.
>
> **Q**: The introduced method is closely related to [1] (InterfaceGAN). More importantly, the qualitative comparison with [1] is missing.
> **A**: Both InterfaceGAN and our method work on finding latent paths that correspond to meaningful semantics. However, we emphasize InterfaceGAN **requires pre-trained classifiers** for every target edit to find their latent paths. This is stated in the Sec. 1 (“... learning classifiers for each desired attribute in the latent space”). In the case when the classifier is missing or hard to train, InterfaceGAN cannot produce a latent path. On the contrary, our method relies on input text and video to find the latent path and can generally work for various motions. Besides our quantitative analysis (Sec. 4.2.1), we add qualitative comparison for baseline comparisons and they are demonstrated in the updated PDF (Figure 14-16). Based on these results, we observe that our method has better disentanglement ability than InterfaceGAN, producing better editing results with much less identity changes.
>
> **Q**: Another closely related work [2] is also not compared.
> **A**: We thank reviewer 3YL2 for suggesting important baseline comparisons. Since this work only releases models for images with 256x256 resolution, we first attempt to train the model on 1024x1024 resolution using the code released by authors for a fair comparison. However, the model does not converge on this resolution. Therefore, we perform comparisons by collecting the source and edited images shown on their papers and using our method to perform the edit. The result is shown in Figure 16, while we believe both methods result in di erent styles of “smile” and “aging”, and the quality is comparable. Also, we emphasize that [2] requires training auxiliary models, while our method does not need any new models, and finding editing direction using our method can be done in a few minutes. Therefore, our method outperforms the listed baselines by providing a more convenient editing framework. For a systematic comparison between our method and baselines, please also refer to the table in common response.
>
> **Q**: The universal directions may not work on GAN models like progressiveGAN. It is questionable if the hypothesis in Fig. 2 can also be generalized to other GAN models.
> **A**: We extend our method on different pretrained GANs in the updated PDF, including BigGAN and progressiveGAN. Our result is demonstrated in Sec. F, Figure 19. From the figure, we observe that both GANs do not synthesize high-quality editing images. For example, for the “opening mouth” attribute in BigGAN, the mouths of dogs in the first two rows are larger, but both the dogs and backgrounds change drastically. This is even worse for the target attribute “closing eyes”. Similarly, in Progressive GAN, we find slight changes toward target attributes “smiling” and “aging”, while the identities are largely changed. This result indicates the latent space in BigGAN and progressive GAN are not highly disentangled. There are two possible reasons: First, the latent code dimension in BigGAN and Progressive GAN (1x512) is smaller than the one in StyleGANv2 (18x512). Second, the hierarchical structure in StyleGAN might lead to better disentanglement. Therefore, compared with these generators, the StyleGANv2 used in the main paper is a better choice.
>
> **Q**: The ablation and discussion on inversion methods are missing. Different inversion methods may affect the performance significantly.
> **A**: We add the ablation study on inversion methods in section G, Figure 20. In this experiment, we consider three inversion methods: Restyle+pSp, Restyle+e4e, and vanilla e4e method. We find that using di fferent inversion methods influences the editing results. Arguably, Restyle+pSp preserves the background color and details best. Besides, we also observe that other methods produce undesired changes (e.g., images are darker for the upper rows). Choosing a faithful inversion method helps produce high-quality edits.
>
> Due to the characters limit, we answer the remaining questions in a separate comment.

---

> ### Author Response · Authors · 2022-12-03
> **Author Response to Reviewer 3YL2 (2/2)**
>
> **Q**: The illustrated micromotion is limited. More complex expressions would be good examples.
> **A**: We add more examples driven by videos in Figure 15. We observe that our method can produce more complex motions ("talking-head" in this example). We also observe our method has a few advantages than baselines in this complex micromotion: (1) Videos synthesized by our method has more significant variance. (2) Our method allows on-demand talking actions, i.e., the synthesized video resembles the reference video at each frame. (3) Our method only requires one reference video rather than a large training dataset.
>
> **Q**: The failure cases and limitations of the proposed method are not provided. It is important to analyze the working scope of the proposed approach to benefit the community.
> **A**: We thank reviewer 3YL2 for suggesting a more complete discussion on failure cases. We have included this discussion in the revised PDF, Sec. H. Specifically, we discuss two failure cases: First, when the pretrained GAN has difficulties in generating images with certain attributes (possibly due to the training dataset, or the limits in GAN architecture), our method cannot produce high-quality editing results. Second, when transferring the editing directions to out-of-domain images, we first need to invert the input images to vectors in the latent space of StyleGAN. When the input largely deviates from a photo-realistic person, the inversion model fails to find the corresponding latent code, and therefore the editing will also fail. We provide two examples in Sec. H. Here, the target attribute is “smiling”', and the input images are anime characters (Mario, Chihiro). We find that the latent code produced by the encoder cannot reconstruct the images (in the second and fifth row), and therefore the editing images have poor quality.
>
> **Q**: Broader Impact Concerns: The face editing technic may have deepfake issues. A Broader Impact Statement is required.
> **A**: We thank reviewer 3YL2 for suggesting more discussions on broader impact. We add broader impact discussions of our method in the revised PDF, section I. We acknowledge that facial editing and video synthesis techniques can be harmful if misused. For example, they can be used to forge fake videos containing offenses and misinformation. However, our method does not try to prompt these societal consequences, but to develop a method that conveniently produces motions for common use. To prevent misuse, our method can be combined with the idea in [1] by adding distinct fingerprints for synthesized videos.
> [1] Yu, Ning, Vladislav Skripniuk, Dingfan Chen, Larry Davis, and Mario Fritz. "Responsible disclosure of generative models using scalable fingerprinting." arXiv preprint arXiv:2012.08726(2020).

---

### Review · Reviewer_LnuT · 2022-11-17

**Summary Of Contributions:**

This paper proposes to use a sequence of images depicting the change of a particular attribute, either using StyleCLIP or real video, and then obtain the PCA of the codes corresponding to the sequence, and then apply these component vectors to other images to achieve desired attribute changes in those images.


**Audience:**

No

**Broader Impact Concerns:**

None.

**Claims And Evidence:**

Yes

**Requested Changes:**

I currently feel that the list of issues I pointed out might be too great to address in a revision. Unless I missed something critical, the method itself seems to be the main problem.

**Strengths And Weaknesses:**

Overall the paper makes sense but I do not think the ideas or observations are new.

The paper lists three contributions so I will try to comment on each one:

> "Leveraging the low-dimensional feature space hypothesis, we demonstrate the properties of StyleGAN’s latent space from a global and universal viewpoint, using “micromotions” as the subject."

I do not understand this contribution at all. What does it mean to demonstrate something "from a global and universal viewpoint"? And what does it mean to demonstrate the properties of a latent space "using 'micromotions' as the subject"?

> "We demonstrate that by using text/video-based anchors, low-dimensional micromotion subspace along with universal and highly disentangled editing directions can be consistently discovered using the same robust subspace projection technique for a large range of micromotion-style facial transformations."

I feel that this contribution is covered by (at least) "GANSpace: Discovering Interpretable GAN Controls". That paper demonstrates that PCA on the latent codes of GANs yields semantically meaningful directions of variation. That paper does it without text-based or video-based anchors, which is maybe even better. It is not surprising that specific factors of variation can be acquired from controlled data.

> "We show the editing direction can be found using a single query face input and then directly applied to other faces, even from vastly different domains (e.g., oil painting, cartoon, and sculpture faces), in an easily controllable way as simple as linear scaling along the discovered subspace."

I am not actually sure what is meant by "using a single query face input" in this sentence, because the method and experiments show the editing direction being obtained from a sequence of images. As for the transfer to other faces, this was again shown already in GANSpace. In general the good transfer of such editing directions is a known phenomenon, often called vector/latent-space arithmetic; for example this is discussed specifically for stylegan2 in "Navigating the GAN Parameter Space for Semantic Image Editing" (CVPR 2021), "StyleFlow: Attribute-conditioned Exploration of StyleGAN-Generated Images using Conditional Continuous Normalizing Flows" (Siggraph 2021).


Results-wise: The micromotions illustrated in Figure 4 seem to be conflating at least two changes at a time. Closed/open eyes is mixed with no-beard/beard. Young/old is mixed with no-beard/beard and also with smiling/neutral.

Method-wise: The method seems to be basically the same as GANspace: PCA on the latent codes, and then walk linearly in the direction of components that control attributes of interest. One important technical difference here is that the current work assumes access to a video sequence of a subject depicting a desired attribute change, and it makes sense that exploiting this will lead to a more accurate direction vector. The text suggests that the main flaw of prior work is "the editing directions founded by these methods are shown to be still entangled with other attributes", but as displayed in Figure 4, this weakness is present in the current work as well.

Presentation-wise: The paper text is riddled with errors. Here are some specific problems and suggested corrections:

Our codes -> Our code

Previous researches -> Previous research

founded -> found [multiple instances of this error]

the following question: whether... -> [this is not a question]

heads movements -> head movements

a series of micromotion -> a series of micromotions

The StyleGAN -> StyleGAN

targeting on image -> targeting image

synthesis task -> synthesis tasks

feature transformation -> feature transformations

Besides -> [in both instances, this word should simply be deleted]

sample-agnostic editing direction -> sample-agnostic editing directions

mild to wild -> mild to extreme

two folds -> two parts

complicacy -> ?? [I don't know what was meant here]

Grasping the Arrow of Time from the Singularity -> ?? [This half of the title has very little to do with the contributions of the work.]

---

> ### Author Response · Authors · 2022-12-03
> **Author Response to Reviewer LnuT (1/2)**
>
> We thank reviewer LnuT for all suggestions.
>
> **Q**: Overall the paper makes sense but I do not think the ideas or observations are new. The paper lists three contributions so I will try to comment on each one.
> * "Leveraging the low-dimensional feature space hypothesis, we demonstrate the properties of StyleGAN’s latent space from a global and universal viewpoint, using “micromotions” as the subject."
> * **Q1**: I do not understand this contribution at all. What does it mean to demonstrate something "from a global and universal viewpoint"? And what does it mean to demonstrate the properties of a latent space "using 'micromotions' as the subject"?
> * **A1**: The “**global and universal**” means our editing can be applied universally to novel images. Specifically, our method boils down to finding meaningful editing directions in the low-dimensional latent space, and we emphasize that the discovered directions can be directly used for novel images. “**using 'micromotions' as the subject**” means we demonstrate the benefits of our method by using synthesized micromotions. As shown in the common response, although other methods (e.g. GANspace, StyleGAN) are able to find similar editing directions, they either rely on auxiliary classifiers or heuristic human choices, which make them less applicable for arbitrary micromotions synthesis. We also add a qualitative comparison in Figure 14 to show the edit quality difference. Our method contributes by synthesizing high-quality micromotions in an automatic and convenient way.
>
> * "We demonstrate that by using text/video-based anchors, low-dimensional micromotion subspace along with universal and highly disentangled editing directions can be consistently discovered using the same robust subspace projection technique for a large range of micromotion-style facial transformations."
> * **Q2**: I feel that this contribution is covered by (at least) "GANSpace: Discovering Interpretable GAN Controls".
> * **A2**: We respectfully disagree. We would like to highlight two differences between GANspace and our method:
> * **(1) In methodology**: GANSpace yields principal directions from randomly sampled latent codes. This means they have no guarantee that the directions contain target semantics, and they have to **inspect each of these components and manually** determine which principal component they want. This is shown in Figure 17, Sec. D. On the contrary, by using the anchoring methods in our work, all the latent codes contain the target semantics, which means we can perform **on-demand disentanglement** instead of picking these semantic aligned directions from a large pool.
> * **(2) In experiment**: Besides, their editing directions are often correlated with other semantics. This phenomenon is discussed in Sec. 4.2, Figure 6, Table 1, and Figure 14. We show that their editing are entangled and hard to transfer, where our method generates better results.
>
> * "We show the editing direction can be found using a single query face input and then directly applied to other faces, even from vastly different domains (e.g., oil painting, cartoon, and sculpture faces), in an easily controllable way as simple as linear scaling along the discovered subspace."
> * **Q3.1**: I am not actually sure what is meant by "using a single query face input" in this sentence, because the method and experiments show the editing direction being obtained from a sequence of images.
> * **A3.1**: We respectfully disagree. In our method, the "single query face input" means our method can work even leveraging input from only one single **identity**. This has been stated in Sec. 3.3: "(a) collecting anchor latent codes from a single identity". And we emphasize that we don't require a series of images as inputs in the text-based method. For the text-based methods, we start from one single image and use StyleCLIP to synthesize a sequence of images.
> * **Q3.2**: As for the transfer to other faces, this was again shown already in GANSpace. In general the good transfer of such editing directions is a known phenomenon, often called vector/latent-space arithmetic; for example this is discussed specifically for stylegan2 in "Navigating the GAN Parameter Space for Semantic Image Editing" (CVPR 2021), "StyleFlow: Attribute-conditioned Exploration of StyleGAN-Generated Images using Conditional Continuous Normalizing Flows" (Siggraph 2021).
> * **A3.2**: While the latent-space arithmetics are explored, we emphasize that the edits produced by current editing methods have difficulties when directly applying them. This is due to their entanglement with other attributes analyzed in Sec. 4.2.1. We compare the listed important baselines in Figure 14-16. We did not include comparisons with “Navigating the GAN Parameter Space for Semantic Image Editing” since this is arithmetics in parameter space rather than latent space.
>
> Due to the characters limit, we answer the remaining questions in a separate comment.

---

> ### Author Response · Authors · 2022-12-03
> **Author Response to Reviewer LnuT (2/2)**
>
> **Q**: Results-wise: The micromotions illustrated in Figure 4 seem to be conflating at least two changes at a time. Closed/open eyes is mixed with no-beard/beard. Young/old is mixed with no-beard/beard and also with smiling/neutral.
> **A**: We acknowledge that target attributes cannot be perfectly disentangled by our method. However, our method shows significantly better disentangling results compared with existing baselines. Please refer to Figure 14-17 for the comparisons and analysis.
>
> **Q**: Method-wise: The method seems to be basically the same as GANspace …The text suggests that the main flaw of prior work is "the editing directions founded by these methods are shown to be still entangled with other attributes", but as displayed in Figure 4, this weakness is present in the current work as well.
> **A**: Please refer to A2 for methodology differences between GANspace and our method. Again, to edit a certain attribute, GANspace requires heuristic human choices to select the best principal components, while our method performs on-demand disentanglement. Besides, while the results of our method are not perfect, from the qualitative and quantitative comparison, our method shows better results.
>
> **Presentation**: We thank reviewer LnuT for pointing out these errors. We fix these in the updated PDF, and we will invite native speakers to polish this work.

---

### Review · Reviewer_QRQY · 2022-11-21

**Summary Of Contributions:**

This submission claims the following contributions:
- The authors find the micromotions can be represented by low-dimensional subspace of the latent space of StyleGAN.
- The directions of the micromotions can be found by text prompts or video frames.
- The direction found using a single query image shows the same effect on different images.

**Audience:**

No

**Broader Impact Concerns:**

Ethics for misusing generative models should be discussed.

**Claims And Evidence:**

Yes

**Requested Changes:**

1) Please discuss the difference between this paper and [MoCoGAN-HD].
1) Please discuss the advantages of robust PCA over vanilla PCA on this task.
1) StyleGANs work well for various datasets. Can the proposed method generalize to different datasets?

> StyleGAN-v2 is indeed “secretly” aware of the subject-disentangled feature variations caused by that micromotion.
1) It is not a `secret` but is known to everyone.

**Strengths And Weaknesses:**

Strengths
1) The experiments show that the found directions lead to the similar effect on different images especially across different domains: pictures, paintings, etc.
1) Video-anchored reference generation is a clever way to prepare the images with the same identity.

Weaknesses
1) A similar idea (using a pretrained generator to produce videos with PCA) already exists: [MoCoGAN-HD]
1) The statements and experiments are limited to faces.
1) The text-anchored reference generation is a minor modification of StyleCLIP: a series of `eyes N% closed` instead of `eyes closed`.
1) Details for applying PCA is not clearly described. Which latent codes are being PCA-ed? Random or anchors? If random, how many? What are the coefficients of the losses for robust PCA? How much signal does PCA lose due to approximation?
1) Although Figure 8 compares the method with and without robust PCA, advantages of robust PCA over vanilla PCA [MoCoGAN-HD] are not provided.

[MoCoGAN-HD] Tian et al., A Good Image Generator Is What You Need for High-Resolution Video Synthesis, ICLR 2021

Minor
> The editing directions founded by these methods are shown to be still entangled with other attributes.
1) Please check the typos. E.g., founded -> found

***********

I checked the acceptance criteria for TMLR:
> Are the claims made in the submission supported by accurate, convincing and clear evidence?

The claims are supported by the evidences. But I do not think that the claims are rigorous enough.

> Would some individuals in TMLR's audience be interested in the findings of this paper?

Yes people are interested in producing videos with image generator. But I do not think the findings worth publishing. What advantages do researchers have if we publish this paper on top of [MoCoGAN-HD]?

---

> ### Author Response · Authors · 2022-12-03
> **Author Response to Reviewer QRQY (1/2)**
>
> We thank reviewer QRQY for all the comments.
>
> **Q**: A similar idea (using a pretrained generator to produce videos with PCA) already exists: [MoCoGAN-HD]
> **A**: Although both works present methods to generate videos from pretrained generators, we emphasize that our work has several fundamental benefits from MoCoGAN-HD from the following aspects:
> (1) **Methodology**: MoCoGAN-HD requires a trained motion generator to synthesize target motion. Different from their method, we target directly finding an editing direction in latent space corresponding to target motions. This is a much simpler and more effective method: It does not **introduce auxiliary models**, and it **allows motion strength control** by simple interpolation or extrapolation on latent space. (described in Sec. 3.3, Step 3 of the main paper).
> (2) **Amount of Training Data**: To synthesize videos with desired motion, MoCoGAN-HD requires a large training dataset (e.g. 22,496 clips in VoxColeb of human speech) to train the motion generator. On the contrary, in the video-anchored method, our framework requires only 1 video to synthesize the corresponding motion. Furthermore, our framework only needs single image input in the text-anchored method. (in Sec. 1, “… such latent subspace can be extracted using only a single query image")
> (3) **Control of motions**: Videos synthesized by MoCoGAN-HD are controlled by sampled motion trajectories. Since the variance of trajectories do not have explicit semantic information, they cannot control details of a motion (e.g. opening mouth in certain frames in "talking-head" task). On the other hand, our method is able to control these fine-grained details.
> The empirical comparison can be found in Figure 15. From the results, we highlight that videos synthesized by our method has more significant variance than MoCoGAN-HD. Our synthesized videos show a person talking with mouth and eyes actions, while most frames in MoCoGAN-HD resemble the first frame and have little changes. With these advantages, we conclude that our method is more effective and convenient than MoCoGAN-HD in the talking-head task.
>
> **Q**: The statements and experiments are limited to faces.
> **A**: We have conducted more experiments on motions for wild animals and show the results in Figure 22. We observe that our method can also synthesize micromotions on wild animals, while the quality is not as good as those on human faces. We highlight two drawbacks here. First, the synthesized images change the background as well. Second, the synthesized images sometimes do not reflect a smooth micromotion. We provide one example in the first row. Specifically, we expect the wild animal to gradually close its eyes, while the synthesized images demonstrate a pixel-wise interpolation from open eyes to close eyes. We hypothesize this is due to the AFHQ-wild dataset does not contain wild animals with dierent eyes open degrees. As such, interpolation on the editing direction cannot synthesize animals with eyes half-open, which is hardly seen in the training dataset. We believe that with a high-quality dataset and better-pretrained generator, we can expect better micromotions.
>
> **Q**: The text-anchored reference generation is a minor modification of StyleCLIP: a series of eyes N% closed instead of eyes closed.
> **A**: Our method is different from StyleCLIP in the following aspects: (1) **Method-wise**: The anchoring methods, including text-anchored and video-anchored generation, are only the first step for finding target latent codes. The key idea of this method is to use the latent codes to form a space and find the correct editing direction. On the contrary, StyleCLIP just finds one latent code corresponding to target edits. (2) **Result-wise**: Directly using StyleCLIP to synthesize motions results in artifacts and entangled attributes. This can be observed in the “no PCA” scenario (transferring editing direction from vanilla StyleCLIP) in Figure 18, Sec. E.
>
> **Q**: Details for applying PCA is not clearly described. Which latent codes are being PCA-ed? Random or anchors? If random, how many? What are the coefficients of the losses for robust PCA? How much signal does PCA lose due to approximation?
> **A**: We thank reviewer QRQY for pointing out the unclear issues here. As stated in Sec. 3.3 Step 2, “... each anchoring latent code serves as the row vector of the data matrix”. Here we were trying to state the **anchors** are used for PCA. Since the latent code dimension is 512, the coefficient lambda is set to 1/sqrt(512) = 0.44. With 4 principal components, we compute the variance loss by applying PCA on 100 edits and PCA results in a 14% variance loss on average.
>
> Due to the characters limit, we answer the remaining questions in a separate comment.

---

> ### Author Response · Authors · 2022-12-03
> **Author Response to Reviewer QRQY (2/2)**
>
> **Q**: The advantages of robust PCA over vanilla PCA [MoCoGAN-HD] are not provided.
> **A**: The comparisons are demonstrated in Figure 18. We show Robust PCA yields the best visual results, followed by Vanilla PCA, while without PCA yields result with the worst visual quality. When comparing the results using vanilla PCA with robust PCA, we can observe the former creates more undesired artifacts. For example, in the third column of Figure 18, we observe vanilla PCA create an unwanted artifact around the shoulder of the sculpture, while robust PCA provides a cleaner image. On the other hand, micromotion subspace without PCA decomposition creates images with the worst quality. Most of them have serve distortion and the faces are barely recognizable. The ablation study demonstrates vanilla PCA is insufficient for a noise-free micromotion subspace, while the outlier-aware Robust PCA is a more favorable choice.
>
> **Q**: The editing directions founded by these methods are shown to be still entangled with other attributes.
> **A**: We acknowledge that our method does not produce perfect disentanglement. However, we would like to emphasize that our method has a stronger disentangling ability and has advantages over existing state-of-the-art methods. Quantitatively, in Sec. 4.2.1 (Figure 6), the results generated by our method show a clear diagonal pattern, indicating editings from our method have much better disentanglement ability. Besides, we qualitatively compare these methods and other strong baselines in Figure 14-16 and observe consistently better or comparable edits.
>
> **Q**: The claims are supported by the evidences. But I do not think that the claims are rigorous enough.
> **A**: In the experiment section (Sec. 4), we design a series of experiments to support our claims. To be specific, we list them as follows:
> * Claim 1: “Leveraging the low-dimensional feature space hypothesis, we demonstrate the properties of StyleGAN’s latent space from a global and universal viewpoint, using “micromotions” as the subject.”
>     * This is supported by experiments in 4.2. In this experiment, we demonstrate various examples of videos (videos can be found in the supplementary materials), showing that we can synthesize a series of micromotions produced by the pretrained image generator, StyleGAN-v2.
> * Claim 2: “We demonstrate that by using text/video-based anchors, low-dimensional micromotion subspace along with universal and highly disentangled editing directions can be consistently discovered using the same robust subspace projection technique for a large range of micromotion-style facial transformations.
>     * This is supported by the analysis in 4.2.1. In the analysis, we qualitatively demonstrate that compared with previous methods (InterfaceGAN, GANspace), our method produces better-disentangled edits. In the revised version, we include more qualitative experiments to better demonstrate the better disentanglement phenomenon.
> * Claim 3: “We show the editing direction can be found using a single query face input and then directly applied to other faces, even from vastly different domains (e.g., oil painting, cartoon, and sculpture faces), in an easily controllable way as simple as linear scaling along the discovered subspace.
>     * This is supported by the experiments in 4.3. In this experiment, we show that our method can be applied to various domains listed in the claim.
> * Besides, we thank all suggestions for reviewer QRQY and we further include more evidence and analysis in the revised version (Sec. D - Sec. J).
>
> **Q**: Yes people are interested in producing videos with image generator. But I do not think the findings worth publishing. What advantages do researchers have if we publish this paper on top of [MoCoGAN-HD]?
> **A**: We demonstrate a series of differences between MoCoGAN-HD and our method. Please refer to the first question for more details. Our method shows clear benefits in both methodology and empirical results compared with MoCoGAN-HD and other baselines (stated in the common response), and therefore we believe our method has unique contributions.
>
> **Q**: Broader Impact: Ethics for misusing generative models should be discussed.
> **A**: We thank reviewer QRQY for suggesting discussions on broader impact. We add discussions in the revised PDF, section I. We acknowledge that facial editing and video synthesis techniques can be harmful if misused. For example, they can be used to forge fake videos containing offenses and misinformation. However, our method does not try to prompt these societal consequences, but to develop a method that conveniently produces motions for common use. To prevent misuse, our method can be combined with the idea in [1] by adding distinct fingerprints for synthesized videos.
> [1] Yu, Ning, Vladislav Skripniuk, Dingfan Chen, Larry Davis, and Mario Fritz. "Responsible disclosure of generative models using scalable fingerprinting." arXiv preprint arXiv:2012.08726(2020).

---

### Author Response · Authors · 2022-12-03
**Common Response**

We thank all the reviewers for their thoughtful and constructive feedback. We have updated the revised PDF according to the suggestions, with more discussions and results on Page 20-28 (Sec. D - Sec. J in the appendix).

In this common response, we focus on the questions related to baselines. First, we summarize the differences between these baselines and our methods in the following table. We emphasize two main differences:

(1) At training time, our method is simple and does not need **auxiliary training datasets or classifiers**. All we need is a pretrained generator, along with a pretrained CLIP in the text-based editing scenario or a single video in the video-based editing scenario.

(2) Besides, for other baseline methods (e.g. GANspace), they rely on randomly sampling latent codes and manually picking from their principal directions. As a consequence, many attributes are clustered. This shows another advantage of our method, that **on-demand disentanglement** vs. picking these semantic aligned directions from a large pool.

| Criteria      | Require auxiliary labelled training dataset | Require auxiliary classifiers | Require manually check attributes | Transfer to novel images? | Motion strength control? |
|---------------|---------------------------------------------|-------------------------------|-----------------------------------|---------------------------|--------------------------|
| InterfaceGAN  | N                                           | Y                             | N                                 | N                         | Y                        |
| GANspace      | N                                           | N                             | Y                                 | N                         | Y                        |
| StyleFlow     | N                                           | Y                             | N                                 | N                         | Y                        |
| Zhuang et al. | N                                           | Y                             | N                                 | N                         | Y                        |
| MoCoGAN-HD    | Y                                           | N                             | N                                 | N                         | N                        |
| Ours          | N                                          | N                             | N                                 | Y                         | Y                        |





Empirically, we compare these baselines in Figure 14-16 in supplementary. In this experiment, we test the editing performance of our methods versus the baseline methods qualitatively on different images. As demonstrated in the figure, our method demonstrates comparable or better performance than existing baselines on representative attributes, including “smiling” and “aging” (the common attributes explored by both our work and baselines). Specifically, compared with InterfaceGAN and GANspace, our method faithfully preserves the identities of the edited subjects, while InterfaceGAN often changes the identities, even genders, during the edition. Meanwhile, existing baselines often require extra steps to edit images. For example, InterfaceGAN and StyleFlow sample 500k and 10k images respectively to train extra models; MoCoGAN-HD requires a training dataset with 20k videos, and Zhuang et al. also need to train an auxiliary model with a long training time. In contrast, our method does not need any extra sampled images or training extra models, thus is significantly convenient compared with these existing methods. Meanwhile, to see the benefits of on-demand disentanglement vs. manually picking from principal components, we demonstrate the first 14 principal directions from GANspace in Figure 17. We observe that (1) all of these latent codes cannot open/close people’s eyes, meaning GANspace fails to find editing directions for this attribute even after manually checking 14 directions in this case; (2) some of the editing directions are clustered, e.g., attribute “glasses” is entangled with “age” in C3, and is entangled with “gender” in C9. On the other hand, our method can find editing directions for these attributes without heuristic human choices.

---

### Decision · Action_Editors · 2022-12-23

**Recommendation:** Reject

**Comment:**

All three reviewers recommend rejection. While they acknowledge the interest of the topic studied in this work, the engineering value of some aspects of the work, and the fact that some of their concerns were addressed by the authors' feedback, they remain of the opinion that this submission does not meet the TMLR acceptance bar. The main reason behind this consensus is the lack of sufficient evidence to support the claims made by the paper.

The Action Editor agrees with the reviewers that this work does not meet the TMLR threshold.

**Audience:**

In general, the TMLR audience would be interested in the topic studied in this paper. However, as stated by Reviewer 3YL2, "Using robust-PCA is a good engineering improvement but not provides much inspiration to the community." This also transpires from the reviews and final recommendations of the other reviewers.


**Claims And Evidence:**

The claims made by the authors are empirically evidenced. However:
- Reviewer QRQY remains unconvinced about the rigor of some claims, and argues that "The response does not make the claims more rigorous, i.e., specific. It only links the claims to the experiments."
- Reviewer LnuT argues that some claimed improvements over the baselines are instead replacements of "some assumptions (e.g., human effort in selecting between a few components) with others (e.g., collecting a video that displays a "micromotion" or using CLIP guidance)";
- All three reviewers argue that the benefits of the proposed method over existing ones are not always clearly demonstrated.